# Complex-Valued Autoencoders for Object Discovery

**Sindy Löwe**                                                   *loewe.sindy@gmail.com*
*UvA-Bosch Delta Lab, University of Amsterdam*

**Phillip Lippe**                                                     *p.lippe@uva.nl*
*QUVA Lab, University of Amsterdam*

**Maja Rudolph**                                              *maja.rudolph@us.bosch.com*
*Bosch Center for AI*

**Max Welling**                                                   *m.welling@uva.nl*
*UvA-Bosch Delta Lab, University of Amsterdam*

**Reviewed on OpenReview:** *https://openreview.net/forum?id=1PfcmFTXoa*

## Abstract

Object-centric representations form the basis of human perception, and enable us to reason about the world and to systematically generalize to new settings. Currently, most works on unsupervised object discovery focus on slot-based approaches, which explicitly separate the latent representations of individual objects. While the result is easily interpretable, it usually requires the design of involved architectures. In contrast to this, we propose a comparatively simple approach – the Complex AutoEncoder (CAE) – that creates distributed object-centric representations. Following a coding scheme theorized to underlie object representations in biological neurons, its complex-valued activations represent two messages: their magnitudes express the presence of a feature, while the relative phase differences between neurons express which features should be bound together to create joint object representations. In contrast to previous approaches using complex-valued activations for object discovery, we present a fully unsupervised approach that is trained end-to-end – resulting in significant improvements in performance and efficiency. Further, we show that the CAE achieves competitive or better unsupervised object discovery performance on simple multi-object datasets compared to a state-of-the-art slot-based approach while being up to 100 times faster to train.

## 1 Introduction

Object discovery plays a crucial role in human perception and cognition (Wertheimer, 1922; Koffka, 1935; Köhler, 1967). It allows us to interact seamlessly with our environment, to reason about it, and to generalize systematically to new settings. To achieve this, our brain flexibly and dynamically combines information that is distributed across the network to represent and relate symbol-like entities, such as objects. The open question of how the brain implements these capabilities in a network of relatively fixed connections is known as the binding problem (Greff et al., 2020).

Currently, most work dedicated to solving the binding problem in machine learning focuses on slot-based approaches (Hinton et al., 2018; Burgess et al., 2019; Greff et al., 2019; Locatello et al., 2020; Kipf et al., 2022). Here, the latent representations are explicitly separated into "slots" which learn to represent different objects. These slots are highly interpretable; however, the introduction of a separate object-centric representation module in a model that otherwise does not exhibit object-centric features causes a number of problems. For one, it usually requires the design of involved architectures with iterative procedures, elaborate structural biases, and intricate training schemes to achieve a good separation of object features into slots. Moreover,

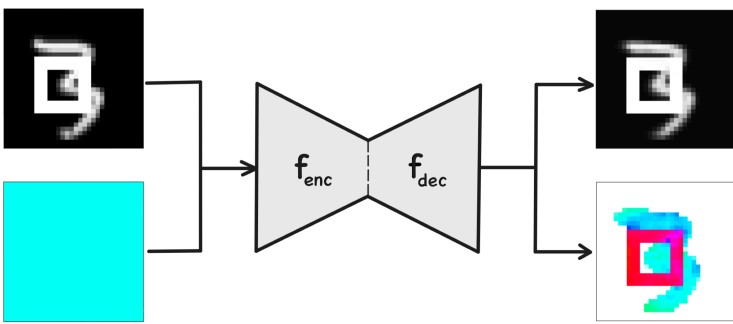

Figure 1: We propose the Complex AutoEncoder (CAE) – a simple and efficient object discovery approach leveraging complex-valued activations in an autoencoding architecture. Given a complex-valued input whose magnitude represents the input image (top left) and whose phase is set to a fixed value (bottom left), the model is trained to reconstruct the input image (top right), and learns to represent the disentangled object identities in its phase values without supervision (bottom right).

this separation is often achieved by limiting the information flow and expressiveness of the model, leading to failure cases for complex objects, e.g. with textured surfaces (Karazija et al., 2021). Finally, since all slots are created at the same level of representation, this approach cannot inherently represent part-whole hierarchies.

To overcome these issues of slot-based approaches, we take inspiration from the temporal correlation hypothesis from neuroscience (Singer & Gray, 1995; Singer, 2009) and design a model that learns representations of objects that are distributed across and embedded in the entire architecture. The temporal correlation hypothesis describes a coding scheme that biological neurons are theorized to use to overcome the binding problem. Essentially, it posits that each neuron sends two types of messages: (1) whether a certain feature is present or not, encoded by the discharge frequency or rate code, and (2) which other neurons to bind information to, encoded by the synchronicity of firing patterns.

Following Reichert & Serre (2014), we abstract away these two messages requiring binary spikes and temporal dynamics by making use of complex-valued activations in artificial neural networks. This allows each neuron to represent the presence of a feature through the complex number's magnitude, and to bind this feature to other neurons' features by aligning its phase value with theirs. After training a Deep Boltzmann Machine (DBM) with real-valued activations, Reichert & Serre (2014) apply this coding scheme at test-time to create object-centric representations. In our experiments, we show that this approach is slow to train due to its greedy layerwise training with Contrastive Divergence (Hinton, 2012), that it is slow to evaluate as it requires 100s-1000s of iterations to settle to an output configuration, and that it leads to unreliable results: even after training the DBM successfully as indicated by its reconstruction performance, its created phase values may not be representative of object identity.

To overcome these limitations of existing complex-valued approaches for object discovery, we propose a convolutional autoencoding architecture – the Complex AutoEncoder (CAE, Fig. 1) – that is trained end-to-end with complex-valued activations. First, we introduce several new, but simple, operations that enable each layer to learn to explicitly control the phase shift that it applies to its features. Then, we train the model in a fully unsupervised way by using a standard mean squared error between its real-valued input and the magnitudes of its complex-valued reconstructions. Interestingly, this simple setup suffices to reliably create phase values representative of object identity in the CAE. Our contributions are as follows:

- We propose the Complex AutoEncoder (CAE), a convolutional architecture that takes inspiration from the neuroscientific temporal correlation hypothesis to create distributed object-centric features.

- We show that the CAE achieves competitive or better object discovery performance on simple, grayscale multi-object datasets compared to SlotAttention (Locatello et al., 2020), a state-of-the-art slot-based approach, while being 10-100 times faster to train.

- We show that the CAE achieves significant improvements in performance and efficiency over the DBM approach proposed by Reichert & Serre (2014).

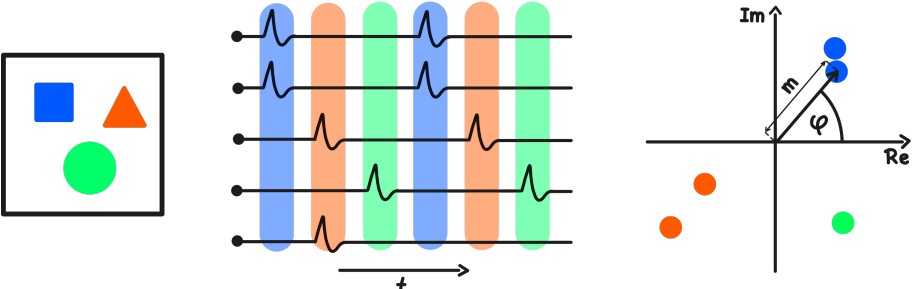

Figure 2: The temporal correlation hypothesis. **Left:** Input image with different objects. **Middle:** Implementation of the temporal correlation hypothesis with spiking neurons. Their spiking rate represents the presence of a feature, while their synchronization represents which features should be bound together to jointly represent an object. **Right:** Implementation of the temporal correlation hypothesis with complex-valued activations. Each complex number $z = m \cdot e^{i\varphi} \in \mathbb{C}$ is defined by its magnitude $m$ and phase $\varphi$. This allows for an equivalent representation of feature presence and synchronization through the magnitude and phase values, respectively.

## 2  The Temporal Correlation Hypothesis

The Complex AutoEncoder takes inspiration from neuroscience, where the temporal correlation hypothesis describes a possible mechanism underlying object-centric representations in the brain. In this section, we will outline this hypothesis, and draw a connection to the complex-valued activations implemented in our proposed model.

In neuroscience, the binding problem describes the open question of how the brain binds information flexibly and dynamically within a network of fixed connectivity to give rise to coherent percepts, e.g. for different objects. Only by overcoming the binding problem, the brain is capable to represent all manner of objects, to attain a compositional understanding of the world, and to generalize robustly to new environments. While there is an ongoing debate as to their functional importance (Shadlen & Movshon, 1999; Ray & Maunsell, 2010), various works posit that the brain uses temporal dynamics to overcome the binding problem (Milner, 1974; Von Der Malsburg & Schneider, 1986; Singer & Gray, 1995; Engel et al., 1997; Singer, 1999; Fries, 2005; Singer, 2009; Fries, 2015; Palmigiano et al., 2017). Essentially, these theories postulate that the brain binds information from different neurons by synchronizing their firing patterns, while desynchronized firing represents information that ought to be processed separately. There are various manifestations of this theory; in this work, we will focus on the temporal correlation hypothesis (Singer & Gray, 1995; Singer, 2009).

The temporal correlation hypothesis describes how the oscillating behavior of biological neurons (a.k.a. brain waves) could be leveraged to overcome the binding problem. It posits that each neuron sends two messages through its spiking pattern (Fig. 2 - Middle): (1) The discharge frequency or rate code of a neuron encodes whether the feature that it is tuned to is present or not. The real-valued activation of neurons in artificial neural networks can be interpreted as the technical implementation of this message. (2) The relative timing between two neurons' spikes encodes whether the represented features of these neurons should be bound together or not. When firing in synchrony, the features they represent will be evaluated jointly by the target neuron and are thus combined in a flexible and dynamic way. Currently, very few works explore the use of this second message type in artificial neural networks.

In this paper, we take inspiration from the temporal correlation hypothesis to develop a machine learning approach capable of overcoming the binding problem. Inspired by previous work (Reichert & Serre, 2014), we abstract away from the spiking nature of biological neural networks and instead represent the two message types described above with the help of complex numbers (Fig. 2 - Right). As a result, we create an artificial neural network with complex-valued activations $z = m \cdot e^{i\varphi} \in \mathbb{C}$ in which the magnitude $m$ can be interpreted as the rate code emitted by a spiking neuron (message (1) above) and the phase $\varphi$ can be used as the mathematical mechanism to capture the temporal alignment of the firing patterns (message (2) above). In the next section, we describe how we implement this coding scheme.

# 3 Complex AutoEncoder

We propose the Complex AutoEncoder (CAE) – an object discovery model that leverages mechanisms inspired by the temporal correlation hypothesis to create distributed object-centric representations. We start by injecting complex-valued activations into a standard autoencoding architecture (Section 3.1). Ultimately, we want these complex-valued activations to convey two messages: their magnitudes should represent whether a feature is present, and their phase values should represent which features ought to be bound together. In Sections 3.2 and 3.3, we describe the setup of the model that gives rise to this coding scheme. As an added benefit, the resulting architecture is equivariant w.r.t. global rotations – an aspect of the model which we will motivate and outline in Section 3.4. Finally, after unsupervised training on a multi-object dataset, the CAE's phase values represent different objects in a scene. In Section 3.5, we describe how we discretize these phase values to produce object-wise representations, as well as pixel-accurate segmentation masks for evaluation.

## 3.1 Complex-Valued Activations in Autoencoders

To enable an autoencoder to develop object-centric representations, we equip it with complex-valued activations. In this section, we will describe how we translate between the real-valued inputs and outputs used for training the model and the complex-valued activations used for representing object-centric features.

The Complex AutoEncoder (Fig. 1) takes a positive, real-valued input image $\mathbf{x}' \in \mathbb{R}^{h \times w}$ with height $h$ and width $w$ and associates each pixel with an initial phase $\boldsymbol{\varphi} = \mathbf{0} \in \mathbb{R}^{h \times w}$ to create the complex-valued input $\mathbf{x}$ to the model:

$$\mathbf{x} = \mathbf{x}' \circ e^{i\boldsymbol{\varphi}} \quad \in \mathbb{C}^{h \times w} \tag{1}$$

where $\circ$ denotes the Hadamard product. The CAE applies a convolutional encoder $f_{\text{enc}}$ and decoder $f_{\text{dec}}$ with real-valued parameters to this complex-valued input to create a complex-valued reconstruction $\hat{\mathbf{z}} = f_{\text{dec}}(f_{\text{enc}}(\mathbf{x})) \in \mathbb{C}^{h \times w}$. To make use of existing deep learning frameworks, in our implementation, we do not apply layers to their complex-valued inputs directly. Instead, each layer extracts real-valued components (the real and imaginary part, or the magnitude and phase) from its input and processes them separately, before combining the results into a complex-valued output. We will describe this process in more detail in the following section.

We create the real-valued reconstruction $\hat{\mathbf{x}}$ by applying $f_{\text{out}}$, a $1 \times 1$ convolutional layer with a sigmoid activation function, on the magnitudes of the complex-valued output $\hat{\mathbf{z}}$ of the decoder: $\hat{\mathbf{x}} = f_{\text{out}}(|\hat{\mathbf{z}}|) \in \mathbb{R}^{h \times w}$. This allows the model to learn an appropriate scaling and shift of the magnitudes to better match the input values. The model is trained by comparing this reconstruction to the original input image using a mean squared error loss $\mathcal{L} = \text{MSE}(\mathbf{x}', \hat{\mathbf{x}}) \in \mathbb{R}^+$ and by using the resulting gradients to update the model parameters.

Finally, we interpret the phase values $\boldsymbol{\varphi} = \arg(\mathbf{z}) \in [0, 2\pi)^{d_{\text{out}}}$ of the $d_{\text{out}}$-dimensional, complex-valued activations $\mathbf{z} \in \mathbb{C}^{d_{\text{out}}}$ as object assignments – either to extract object-wise representations from the latent space or to obtain a pixel-accurate segmentation mask in the output space. Here, $\arg(\mathbf{z})$ describes the angles between the positive real axis and the lines joining the origin and each element in $\mathbf{z}$.

## 3.2 Phase Alignment of Complex Numbers

For the CAE to accomplish good object discovery performance, the phases of activations representing the same object should be synchronized, while activations induced by different objects should be desynchronized. To achieve this, we need to enable and encourage the network to assign the same phases to some activations and different phases to others, and to precisely control phase shifts throughout the network. We accomplish this by implementing the following three design principles in each network layer $f_{\boldsymbol{\theta}} \in \{f_{\text{enc}}, f_{\text{dec}}\}$ parameterized by $\boldsymbol{\theta} \in \mathbb{R}^k$ (where $k$ represents the number of parameters in that layer) and applied to the $d_{\text{in}}$-dimensional input to that layer $\mathbf{z} \in \mathbb{C}^{d_{\text{in}}}$:

**Synchronization** First, we need to encourage the network to synchronize the phase values of features that should be bound together. This principle is fulfilled naturally when using additive operations between

complex numbers: when adding two complex numbers of opposing phases, they suppress one another or even cancel one another out (a.k.a. destructive interference). Thus, to preserve features, the network needs to align their phase values (a.k.a. constructive interference).

**Desynchronization** Next, we need a mechanism that can desynchronize the phase values. Again, this is achieved naturally when using additive operations between complex numbers: when adding two complex numbers with a phase difference of $90°$, for example, the result will lie in between these two numbers and thus be shifted, i.e. desynchronized by $45°$. On top of this inherent mechanism, we add a second mechanism that lends the network more control over the precise phase shifts. Specifically, we first apply the weights $\mathbf{w} \in \boldsymbol{\theta}$ of each layer separately to the real and imaginary components of its input:

$$\boldsymbol{\psi} = f_{\mathbf{w}}(\mathbf{z}) = f_{\mathbf{w}}(\mathrm{Re}(\mathbf{z})) + f_{\mathbf{w}}(\mathrm{Im}(\mathbf{z})) \cdot i \quad \in \mathbb{C}^{d_{\mathrm{out}}} \tag{2}$$

where $d_{\mathrm{out}}$ represents the output dimensionality of the layer and $\cdot$ denotes scalar multiplication. Then, we add separate biases $\boldsymbol{b_m}, \boldsymbol{b_\varphi} \in \boldsymbol{\theta}$ to the magnitudes and phases of the resulting complex-valued representations $\boldsymbol{\psi}$ to create the intermediate magnitude $\boldsymbol{m_\psi}$ and phase $\boldsymbol{\varphi_\psi}$:

$$\boldsymbol{m_\psi} = |\boldsymbol{\psi}| + \boldsymbol{b_m} \quad \in \mathbb{R}^{d_{\mathrm{out}}} \qquad\qquad \boldsymbol{\varphi_\psi} = \arg(\boldsymbol{\psi}) + \boldsymbol{b_\varphi} \quad \in \mathbb{R}^{d_{\mathrm{out}}} \tag{3}$$

This formulation allows the model to influence the phase value of each activation directly through the bias $\boldsymbol{b_\varphi}$, and thus to learn explicit phase shifts throughout the network. Additionally, the bias $\boldsymbol{b_\varphi}$ allows the model to break the symmetry created by the equal phase initialization (Eq. (1)).

**Gating** Finally, we add a gating mechanism that selectively weakens out-of-phase inputs. We implement this by applying each layer to the magnitude of its input and combine the result with $\boldsymbol{m_\psi}$ to create the intermediate values $\boldsymbol{m_z}$:

$$\begin{aligned} \boldsymbol{\chi} &= f_{\mathbf{w}}(|\mathbf{z}|) + \boldsymbol{b_m} \qquad \in \mathbb{R}^{d_{\mathrm{out}}} \\ \boldsymbol{m_z} &= 0.5 \cdot \boldsymbol{m_\psi} + 0.5 \cdot \boldsymbol{\chi} \quad \in \mathbb{R}^{d_{\mathrm{out}}} \end{aligned} \tag{4}$$

As shown by Reichert & Serre (2014), this implementation effectively masks out inputs that have an opposite phase compared to the overall output. Formally speaking, given a set of input features $\mathbf{z}_1$ with the same phase value and an out-of-phase input feature $z_2$ with a maximally distant phase[1], if $|\langle \mathbf{w_1}, \mathbf{z}_1 \rangle| > |w_2 z_2|$, $\langle \mathbf{w_1}, |\mathbf{z_1}| \rangle > 0$ and $w_2 > 0$, the net contribution of $z_2$ to the output is zero[2]. Besides this hard gating for maximally out-of-phase inputs, this mechanism further leads to progressively softer gating for smaller phase differences. Overall, this allows for the flexible binding of features by determining the effective connectivity between neurons based on the phase values of their activations. Since features with different phase values are processed separately, this ultimately encourages the network to assign different phase values to the features of different objects.

## 3.3 Complex-Valued Activation Function

We propose a new activation function for complex-valued activations to further ensure maximal control of the network over all phase shifts. To create a layer's final output $\mathbf{z}'$, we apply a non-linearity on the intermediate values $\boldsymbol{m_z}$, but keep the phases $\boldsymbol{\varphi_\psi}$ unchanged:

$$\mathbf{z}' = \mathrm{ReLU}(\mathrm{BatchNorm}(\boldsymbol{m_z})) \circ e^{i\boldsymbol{\varphi_\psi}} \quad \in \mathbb{C}^{d_{\mathrm{out}}} \tag{5}$$

There are several things to note about this setup. First, $\boldsymbol{m_z}$ is positive valued, unless the magnitude bias $\boldsymbol{b_m}$ pushes it into the negative domain (Eq. (3), Eq. (4)). Second, by applying BatchNormalization (Ioffe & Szegedy, 2015), we ensure that – at least initially – $\boldsymbol{m_z}$ becomes zero-centered and therefore makes use of the non-linear part of the ReLU activation function (Krizhevsky et al., 2012). At the same time, BatchNormalization provides the flexibility to learn to shift and scale these values if appropriate. Finally, the ReLU non-linearity ensures that the magnitude of $\mathbf{z}'$ is positive and thus prevents any phase flips.

---

[1]We say that $n$ complex numbers have "maximally distant phases" if they maximize their pairwise cosine distance to one another.

[2]Denoting the inner product as $\langle \cdot, \cdot \rangle$ and leaving out the bias $b_m$ for clarity, the proof for this is: $\boldsymbol{m_z} = 0.5 \cdot (|\langle \mathbf{w_1}, \mathbf{z}_1 \rangle + w_2 z_2|) + 0.5 \cdot (\langle \mathbf{w_1}, |\mathbf{z_1}| \rangle + w_2 z_2) = 0.5 \cdot (|\langle \mathbf{w_1}, \mathbf{z}_1 \rangle| - w_2 z_2|) + 0.5 \cdot (\langle \mathbf{w_1}, |\mathbf{z_1}| \rangle + w_2 z_2) = 0.5 \cdot (|\langle \mathbf{w_1}, \mathbf{z}_1 \rangle|) + 0.5 \cdot (\langle \mathbf{w_1}, |\mathbf{z_1}| \rangle)$

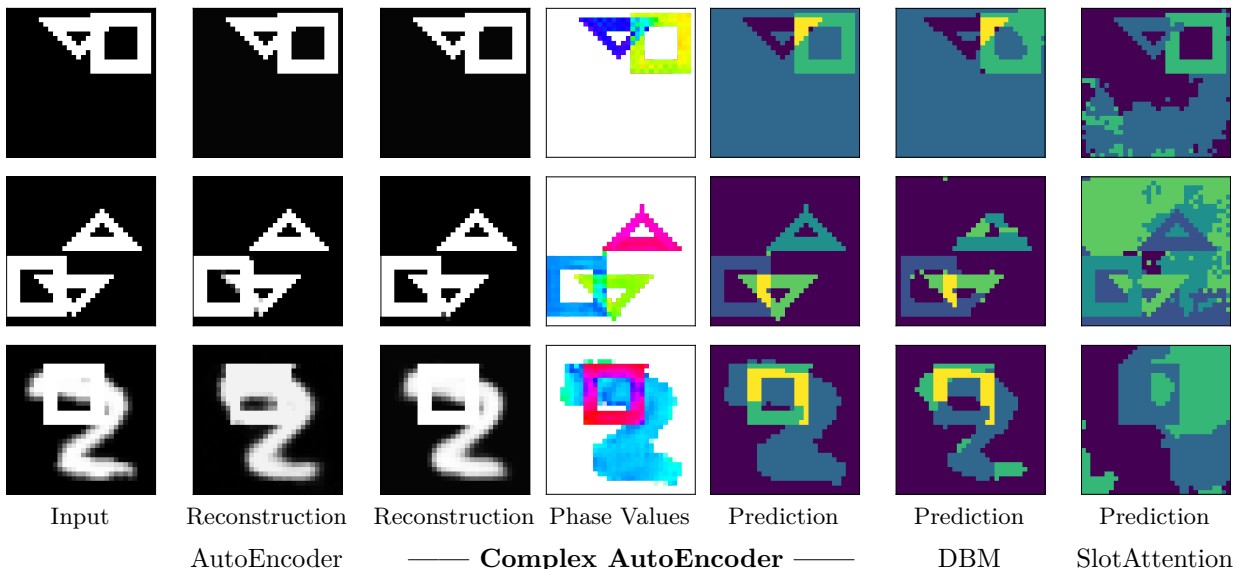

| Input | Reconstruction | Reconstruction | Phase Values | Prediction | Prediction | Prediction |
|---|---|---|---|---|---|---|
| | AutoEncoder | —— **Complex AutoEncoder** —— | | | DBM | SlotAttention |

Figure 3: Visual comparison of the performance of the Complex AutoEncoder, a corresponding real-valued autoencoder, a DBM model and a SlotAttention model on random test-samples of the 2Shapes (**Top**), 3Shapes (**Middle**) and MNIST&Shape (**Bottom**) datasets. Areas in which objects overlap are removed before evaluation, resulting in yellow areas in the predictions of the CAE and the DBM model. The Complex AutoEncoder produces accurate reconstructions and object separations.

### 3.4 Global Rotation Equivariance

Given the setup as described above, the CAE is equivariant with regard to global rotations, i.e. to global phase shifts of the input.

**Proposition 3.1.** *Let $CAE(\mathbf{x}) = f_{dec}(f_{enc}(\mathbf{x})) \in \mathbb{C}^{h \times w}$ be the output of the autoencoding model given the input $\mathbf{x} \in \mathbb{C}^{h \times w}$. Let $\alpha \in [0, 2\pi)$ be an arbitrary phase offset. Then, the following holds:*

$$CAE(\mathbf{x} \cdot (\cos(\alpha) + \sin(\alpha)i)) = CAE(\mathbf{x}) \cdot (\cos(\alpha) + \sin(\alpha)i) \tag{6}$$

*where multiplication with $\cos(\alpha) + \sin(\alpha)i$ corresponds to a counter-clockwise rotation by $\alpha$.*

The proof and empirical validation for this proposition can be found in Appendix D.1. This means that when we shift the phase value associated with the pixels of an input image (e.g. by choosing a different value for $\varphi$ in Eq. (1)), the output phases will be shifted correspondingly, but the output magnitudes will remain unchanged. This property is important when considering the temporal analogy underlying the CAE: the point in time at which a stimulus is perceived (here: input phase) should influence *when* the stimulus is processed (here: output phase), but it should not influence *how* it is processed (here: output magnitudes). Additionally, without this equivariance property, the model could use the absolute phase values to learn to separate objects instead of binding features "by synchrony", i.e. based on their relative phase differences to one another.

### 3.5 Creating Discrete Object Assignments from Continuous Phase Values

To extract object-wise representations from the latent space, as well as pixel-accurate segmentation masks from the output space, we create discrete object assignments for each feature by applying a clustering procedure to the phase values. We start this clustering procedure with two pre-processing steps. First, we account for the circular nature of the phase values by mapping them onto a unit circle. This prevents values close to 0 and $2\pi$ from being assigned to different clusters despite representing similar angles. Then, to account for the fact that the phase values of complex numbers with small magnitudes become increasingly random, we scale features by a factor of $10 \cdot m$ if their corresponding magnitude $m < 0.1$. As a result of this

Table 1: MSE and ARI scores (mean ± standard error across 8 seeds) for the Complex AutoEncoder, its real-valued counterpart (AutoEncoder), a DBM model and a SlotAttention model in three simple multi-object datasets. The proposed Complex AutoEncoder achieves better reconstruction performance than its real-valued counterpart on all three datasets. Additionally, it significantly outperforms the DBM model across all three datasets, and its object discovery performance is competitive to SlotAttention on the 2Shapes and 3Shapes datasets. Finally, the CAE manages to disentangle the objects in the MNIST&Shape dataset, where both DBM and SlotAttention fail.

| Dataset | Model | MSE ↓ | ARI+BG ↑ | ARI-BG ↑ |
|---|---|---|---|---|
| 2Shapes | Complex AutoEncoder | $\mathbf{3.322e\text{-}04}_{\pm\ 1.583e\text{-}06}$ | $\mathbf{0.999}_{\pm\ 0.000}$ | $\mathbf{1.000}_{\pm\ 0.000}$ |
| | AutoEncoder | $5.565e\text{-}04_{\pm\ 2.900e\text{-}04}$ | — | — |
| | DBM | $3.308e\text{-}03_{\pm\ 1.024e\text{-}04}$ | $0.920_{\pm\ 0.002}$ | $0.744_{\pm\ 0.010}$ |
| | SlotAttention | $1.419e\text{-}04_{\pm\ 1.410e\text{-}04^*}$ | $0.812_{\pm\ 0.081}$ | $\mathbf{1.000}_{\pm\ 0.000}$ |
| 3Shapes | Complex AutoEncoder | $\mathbf{1.313e\text{-}04}_{\pm\ 2.020e\text{-}05}$ | $\mathbf{0.976}_{\pm\ 0.002}$ | $\mathbf{1.000}_{\pm\ 0.000}$ |
| | AutoEncoder | $8.568e\text{-}04_{\pm\ 9.878e\text{-}05}$ | — | — |
| | DBM | $1.045e\text{-}02_{\pm\ 1.494e\text{-}04}$ | $0.856_{\pm\ 0.006}$ | $0.419_{\pm\ 0.023}$ |
| | SlotAttention | $1.827e\text{-}04_{\pm\ 3.125e\text{-}05^*}$ | $0.107_{\pm\ 0.008}$ | $0.997_{\pm\ 0.001}$ |
| MNIST&Shape | Complex AutoEncoder | $\mathbf{3.185e\text{-}03}_{\pm\ 1.514e\text{-}04}$ | $\mathbf{0.783}_{\pm\ 0.004}$ | $\mathbf{0.971}_{\pm\ 0.011}$ |
| | AutoEncoder | $5.792e\text{-}03_{\pm\ 5.553e\text{-}04}$ | — | — |
| | DBM | $1.560e\text{-}02_{\pm\ 8.069e\text{-}05^*}$ | $0.718_{\pm\ 0.002}$ | $0.175_{\pm\ 0.006}$ |
| | SlotAttention | $5.438e\text{-}03_{\pm\ 1.607e\text{-}04^*}$ | $0.047_{\pm\ 0.013}$ | $0.089_{\pm\ 0.028}$ |

*The grayed-out performances are not comparable due to the use of different autoencoding setups.

scaling, these features will fall within the unit circle, close to the origin. In our experiments, we find that they tend to be assigned their own cluster and usually represent the background. As outlined in Appendix C.1, this re-scaling of features is crucial to ensure little noise in the evaluation procedure, but currently prevents the CAE from being applied to RGB data. Finally, we apply $k$-means, with $k$ corresponding to the number of objects in the input plus one for the background, and interpret the resulting cluster assignment for each phase value as the predicted object assignment of the corresponding feature.

Note that we could replace $k$-means with any other clustering algorithm to relax the requirement of knowing the number of objects in advance. Additionally, this discretization of object assignments is only required for the evaluation of the CAE. During training, the CAE learns continuous object assignments through its phase values and thus creates the appropriate amount of clusters automatically.

## 4 Results

In this section, we evaluate whether the Complex AutoEncoder can learn to create meaningful phase separations representing different objects in an unsupervised way. We will first describe the general setup of our experiments, before investigating the results across various settings. Our code is publicly available at `https://github.com/loeweX/ComplexAutoEncoder`.

### 4.1 Setup

**Datasets** We evaluate the Complex AutoEncoder on three grayscale datasets: 2Shapes, 3Shapes, and MNIST&Shape. For each of these datasets, we generate 50,000 training images and 10,000 images for validation and testing, respectively. All images contain $32 \times 32$ pixels. The 2Shapes dataset represents the easiest setting, with two randomly placed objects ($\square, \triangle$) in each image. The 3Shapes dataset contains a third randomly placed object ($\triangledown$) per image. This creates a slightly more complex setting due to the higher object count, the two similar shapes ($\triangle, \triangledown$), and stronger overlap between objects. Finally, the MNIST&Shape dataset combines an MNIST digit (LeCun et al., 2010) and a randomly placed shape ($\square$ or $\triangledown$) in each image. This creates a challenging setting with more diverse objects. Finally, for evaluation, we generate pixel-accurate segmentation masks for all images. More details in Appendix C.4.

**Model & Training**  We make use of a fairly standard convolutional autoencoder architecture, as presented in Lippe (2021) (details in Appendix C.1). We train the model using Adam (Kingma & Ba, 2015) and a batch-size of 64 for 10,000 – 100,000 steps depending on the dataset. Within the first 500 steps of training, we linearly warm up the learning rate (Goyal et al., 2017) to its final value of $1e-3$. All experiments are implemented in PyTorch 1.10 (Paszke et al., 2019) and were run on a single Nvidia GTX 1080Ti. To ensure the comparability of runtimes between models, all experiments were run on the same machine and with the same underlying implementations for data-loading, training and evaluation wherever possible.

**Baselines**  We compare the CAE to three baseline models. First, we compare it against a Deep Boltzmann Machine (DBM; Salakhutdinov & Hinton (2009)) that closely follows Reichert & Serre (2014). We test two architectures for this model, one resembling the setup of the CAE (6-layer DBM) and one following the architecture used by Reichert & Serre (2014) (3-layer DBM), and report the results of the best performing model for each dataset. For a full breakdown of all results, see Appendix D.2. Additionally, we provide an in-depth discussion on the limitations of our re-implementation in Appendix C.2. The second baseline we consider is a state-of-the-art slot-based approach: SlotAttention (Locatello et al., 2020). SlotAttention is an iterative attention mechanism that produces $k$ slots which learn to represent individual objects in the input. It has achieved remarkable unsupervised object discovery results on synthetic multi-object datasets, while being more memory efficient and faster to train than other slot-based approaches. For more details, see Appendix C.3. Finally, we compare the CAE against a corresponding real-valued autoencoder. This model uses the same general architecture and training procedure as the CAE, but does not employ complex-valued activations or any of the mechanisms described in Section 3.2. It does, however, apply BatchNormalization before each ReLU as we have found that this improves performance in the real domain as well.

**Metrics**  We use three metrics to evaluate the performance of the CAE and to compare it with the baselines. We measure the reconstruction performance in terms of mean squared error (MSE). To assess the object discovery performance, we compute Adjusted Rand Index (ARI) scores (Rand, 1971; Hubert & Arabie, 1985). ARI measures clustering similarity, where a score of 0 indicates chance level and a score of 1 indicates a perfect match. We utilize this metric in two ways. First, following previous work (Greff et al., 2019; Locatello et al., 2020), we evaluate "ARI-BG" where we exclude the background labels from the evaluation. Additionally, we assess "ARI+BG" which evaluates the performance on all pixels. One thing to note here is that a model can achieve relatively high ARI+BG scores by assigning all pixels to the same label – thus, this score is only meaningful in conjunction with a high ARI-BG score to ensure good object separation. For both ARI scores, we remove areas in which objects overlap from the evaluation, as they are ambiguous in the grayscale setting.

## 4.2 Evaluation

First, we compare the quantitative performance of the CAE against the three baselines in Table 1.

**Object Discovery Performance**  The *Complex AutoEncoder* achieves considerably better object discovery performance than the *DBM* model across all three tested datasets. In fact, we find that our re-implementation of the DBM model only achieves consistent object separation in the simplest 2Shapes dataset, and largely fails on the other two – despite achieving a reasonable reconstruction performance on all datasets (see Fig. 7 in the Appendix). When comparing the object discovery performance of the CAE against *SlotAttention*, we make three observations: (1) Both models achieve (near) perfect ARI-BG scores on the 2Shapes and 3Shapes datasets. (2) On all datasets, the CAE achieves considerably better ARI+BG scores indicating a more accurate separation of foreground and background. (3) On the MNIST&Shape dataset, the CAE achieves an almost perfect ARI-BG score, while SlotAttention's performance is close to chance level (we provide a detailed analysis of this failure mode of SlotAttention in Appendix C.3). Overall, this shows that the Complex AutoEncoder achieves strong object discovery performance on the three datasets considered. On top of this, despite its simple and efficient design, it can overcome the challenges set by the MNIST&Shape dataset (high diversity in object shapes and relatively large object sizes), while neither the DBM model nor SlotAttention can.

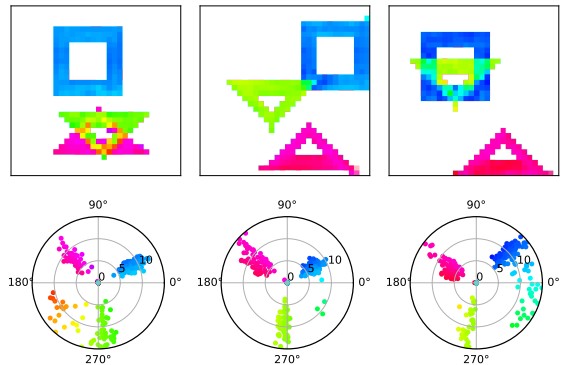 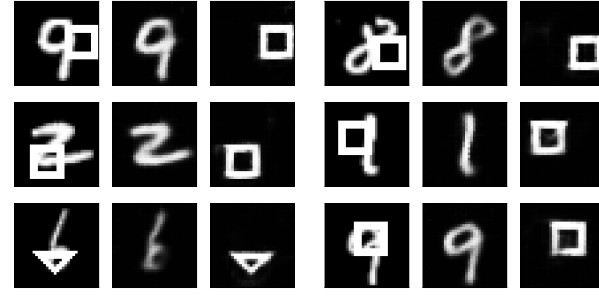

Figure 4: Phase separation in the CAE. **Top**: Output phase images. **Bottom**: Plotting every output value in the complex plane and applying the same color coding as above. The cluster centroids of the phase values belonging to the individual objects have almost maximally distant phases. Interestingly, areas in which the objects overlap get assigned intermediate phase values.

Figure 5: Investigating object-centricity of the latent features in the CAE. **Columns 1 & 4**: input images. **Columns 2-3 & 5-6**: object-wise reconstructions. By clustering features created by the encoder according to their phase values, we can extract representations of the individual objects and reconstruct them separately.

**Reconstruction Performance** The *Complex AutoEncoder* creates more accurate reconstructions than its real-valued counterpart (*AutoEncoder*) on all three multi-object datasets. This illustrates that the CAE uses its complex-valued activations to create a better disentanglement of object features.

**Object-Centric Representations** To evaluate whether the CAE creates object-centric representations throughout the model, we cluster the latent features created by the encoder (Section 3.5), and fine-tune the decoder to reconstruct separate objects from the individual clusters. As shown in Fig. 5, this allows us to create accurate reconstructions of the individual objects. We conclude that the latent representations of our model are object-centric: the representations of the individual objects can be clearly separated from one another based on their phase values, and they contain all the necessary information for the individual objects to be reconstructed accurately.

**Qualitative Evaluation** In Fig. 3, we show exemplary outputs of the four compared models for each dataset (for more results, see Appendix D.6). When looking more closely at the phase separation created by the Complex AutoEncoder as shown in Fig. 4, we find that it assigns almost maximally distant phase values to the different objects. Interestingly, the phase values of overlapping areas tend to fall in between the phase values of the individual objects. Since it is ambiguous in the 3Shapes dataset, which object is in the foreground and which one is in the background, this shows that the model accurately expresses the uncertainty that it encounters in these overlapping areas.

**Dataset Sensitivity Analysis** We explore the capabilities and limitations of the CAE in different dataset settings. To achieve this, we generate several variations of the grayscale datasets considered before. For a detailed description of the datasets considered, as well as a full breakdown of the results, see Appendix D.4.

*Variable backgrounds.* We implement a version of the 2Shapes dataset in which we randomly sample the background color of each image. Since the CAE achieves an ARI-BG score of $0.995 \pm 0.001$, we conclude that it can handle images with variable backgrounds.

*Duplicate objects.* We assess the CAE on a dataset in which each image contains two triangles of the same orientation. The CAE achieves an ARI-BG score of $0.948 \pm 0.045$ here, indicating that it can handle multiple instances of the same object within an image.

*Object count.* To investigate whether the CAE can represent a larger number of objects, we construct a dataset with four objects per image. This builds on the 3Shapes dataset, but adds an additional circle to each image. On this dataset, the performance of the CAE drops substantially to an ARI-BG score of $0.669 \pm 0.029$. To distinguish whether this lowered performance is due to the larger object count per image or due to the larger variability of objects, we implemented a dataset with five different shapes (four of which are the same as in the previous dataset, plus an additional larger circle) out of which two are randomly sampled for each image. Here, the CAE achieves an ARI-BG score of $0.969 \pm 0.007$ – highlighting that it is mostly restricted in the number of objects it can represent at once, but that it is able to represent a larger range of object types across images.

*Generalization.* We test whether the CAE generalizes to different object counts. For this, we train the model on the 2Shapes dataset and test it on a dataset in which each image contains 3 shapes (one square and two triangles of the same orientation). We find that the performance drops substantially in this setting, with the model achieving an ARI-BG score of $0.758 \pm 0.007$. Thus, it seems that the CAE does not generalize to more objects than observed during training. It does, however, generalize to settings in which fewer objects are present: after training the CAE on the 3Shapes dataset, its performance remains high at $0.956 \pm 0.019$ ARI-BG when tested on a variation of this dataset where images may contain either one, two or three objects.

**Model Sensitivity Analysis**  We evaluate the influence of certain design decisions on the performance of the Complex AutoEncoder in Table 2. We find that without the phase-bias $b_\varphi$ (Eq. (3)), the model is unable to push its activations off the real-axis and therefore achieves no meaningful phase separation between objects; without the $\chi$ term (Eq. (4)), activations are not gated according to their phases and the model has no incentive to separate the object representations; and without BatchNormalization (Eq. (5)), the network cannot make appropriate use of the non-linear part of the ReLU leading to inferior performance. In all three cases, this results in the CAE failing to achieve

Table 2: Model sensitivity analysis on the 2Shapes dataset (mean $\pm$ standard error across 8 seeds). We find that there are several crucial components contributing to the CAE's ability to achieve meaningful phase separation without supervision.

| Name | MSE | ARI-BG |
|---|---|---|
| Complex AutoEncoder | 3.322e-04 $_{\pm\ 1.583\text{e-}06}$ | 1.000 $_{\pm\ 0.000}$ |
| $- b_\varphi$ | 5.127e-04 $_{\pm\ 6.793\text{e-}05}$ | 0.100 $_{\pm\ 0.035}$ |
| $- \chi$ | 3.227e-03 $_{\pm\ 5.013\text{e-}04}$ | 0.074 $_{\pm\ 0.068}$ |
| $-$ BatchNorm | 6.165e-02 $_{\pm\ 2.228\text{e-}02}$ | 0.373 $_{\pm\ 0.137}$ |
| $- f_{\text{out}}$ | 2.462e-03 $_{\pm\ 1.458\text{e-}03}$ | 0.939 $_{\pm\ 0.039}$ |

any meaningful object separation. While the functionality that each of these components provides is crucial for the CAE's performance, alternative approaches implementing similar functionalities may be possible. Our results further indicate that the final $1 \times 1$ convolutional layer $f_{\text{out}}$ that creates the real-valued reconstructions from the magnitudes of the complex-valued outputs improves the model's performance. Finally, we observe that the bottleneck size has relatively little influence on the performance of the Complex AutoEncoder (Fig. 9 in the Appendix). This indicates that the CAE does not need to restrict the expressivity of the model to create disentangled object representations.

**Global Rotation Equivariance**  We visually highlight the global rotation equivariance property of the CAE in Fig. 6. As expected, a shift in the input phase value leads to a corresponding shift in the output phases, but leaves the output magnitudes unchanged.

**Runtime**  Due to its non-iterative design and its end-to-end optimization with comparatively few training steps, the CAE is between 1.1 and 21.0 times faster to train than the DBM model, and between 10.0 and 99.8 times faster than SlotAttention. The precise values depend on the dataset and DBM architecture used. On the 2Shapes dataset, for example, the CAE is 2.1 times faster to train than the DBM model and 99.8 times faster than SlotAttention. Besides this, we find that the discretization of phase values with $k$-means leads to significantly slower evaluation times for both the CAE and DBM. As a result, the CAE is approximately 10 times slower to evaluate than SlotAttention. The DBM is slowed down even further by its iterative settling procedure, leading to 2.0 - 16.6 times slower evaluation times compared to the CAE. See Appendix D.5 for a detailed breakdown of all runtimes.

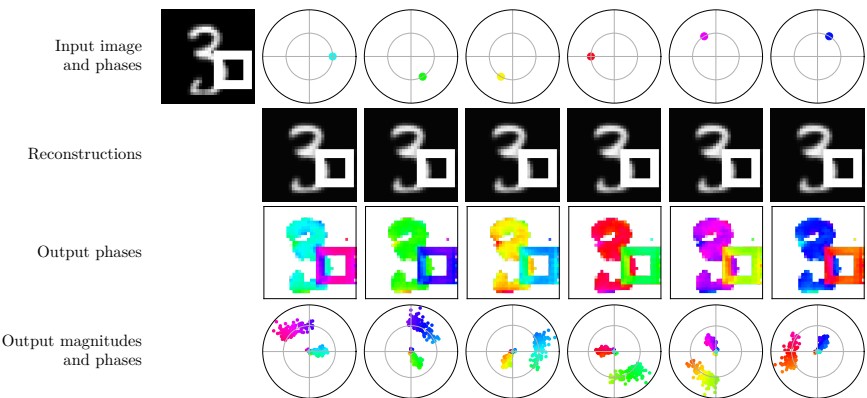

Figure 6: Visual evaluation of the global rotation equivariance of the CAE. Given an input with different phase values (**1st row**), the output magnitudes (and therefore the reconstructions) remain unchanged (**2nd/4th row**), but the output phase values shift correspondingly (**3rd/4th row**).

## 5 Related Work

**Object Discovery**   There is a broad range of approaches attempting to solve the binding problem in artificial neural networks (see Greff et al. (2020) for a great overview). However, most works focus on one particular representational format for objects: slots. These slots create an explicit separation of the latent representations for different objects. Additionally, they create an explicit separation between a specialized object-centric representation part of the model and non-object-centric parts that merely support the former, for example, through encoding and decoding functionality. In contrast to this, the Complex AutoEncoder embeds object-centricity into the entire architecture and creates distributed object-centric representations.

To create slot-based object-centric representations, different mechanisms have been proposed to overcome the binding problem, i.e. to break the symmetry between the representations of different objects in a network of fixed weights. One way to break this symmetry is to enforce an order along a certain axis. This can be achieved by imposing an order on the slots (Eslami et al., 2016; Burgess et al., 2019; Engelcke et al., 2020), by assigning slots to particular spatial coordinates (Santoro et al., 2017; Crawford & Pineau, 2019; Lin et al., 2020), or by learning specialized slots for different object types (Hinton et al., 2011; 2018). Approaches that create the most general slot representations do not enforce any such order, but require iterative procedures to break the symmetries instead (Greff et al., 2016; 2017; 2019; Goyal et al., 2021; Kosiorek et al., 2020; Stelzner et al., 2021; Du et al., 2021). SlotAttention (Locatello et al., 2020), which falls into this final category, breaks the symmetry between slots through an iterative attention mechanism. Many recent works make use of SlotAttention: Kipf et al. (2022); Singh et al. (2022); Elsayed et al. (2022) extend it to video data, Singh et al. (2021) propose an image generation model utilizing SlotAttention; and Löwe et al. (2020) propose a self-supervised training approach for SlotAttention.

**Object Discovery with Complex-Valued Networks**   A variety of research has explored different activation functions, training regimes, and applications for complex-valued neural networks (see Bassey et al. (2021) for a review). Despite this, there has been little research on the use of complex-valued networks for object discovery. The earliest works in this direction are by Mozer et al. (1992); Zemel et al. (1995). Their architectures learn to assign different phase values to different objects through a supervised training procedure. Rao et al. (2008); Rao & Cecchi (2010; 2011) enable their complex-valued neural networks to separate overlapping objects on test images by training them on images of individual objects of the same type. Finally, the only existing complex-valued network for object discovery that is fully unsupervised was developed by Reichert & Serre (2014). They train a real-valued Deep Boltzmann Machine on datasets similar to the ones presented here. At test time, they inject complex-valued activations to create phases representative of object identity.

All these methods initialize the complex-valued inputs to their networks with the magnitudes of the input images and with random phase values. As a result, they require iterative procedures with 10s-1000s of iterations to settle to an output configuration. In contrast to that, the proposed Complex AutoEncoder initializes all phase values with a fixed value and – after fully unsupervised, end-to-end training with complex-valued activations – only requires a single forward-pass through the model. Further, the CAE is the first complex-valued network for object discovery that has been shown to be equivariant to global rotations. All together, this greatly improves the efficiency and performance of complex-valued neural networks for object discovery.

## 6 Conclusion

**Summary**  We present the Complex AutoEncoder – an object discovery approach that takes inspiration from neuroscience to implement distributed object-centric representations. After introducing complex-valued activations into a convolutional autoencoder, it learns to encode feature information in the activations' magnitudes and object affiliation in their phase values. We show that this simple and fully unsupervised setup suffices to achieve strong object discovery results on simple multi-object datasets while being fast to train.

**Limitations and Future Work**  The proposed Complex AutoEncoder constitutes a first step towards efficient distributed object-centric representation learning, but some limitations remain. Most importantly, SlotAttention and other slot-based approaches are generally applicable to more challenging datasets: they can handle RGB inputs, for example, and images with a larger number of objects. Nonetheless, the Complex AutoEncoder provides an important step forward by proposing a simple and efficient non-iterative design that considerably outperforms previous complex-valued approaches and, for the first time, was shown to achieve competitive results to a slot-based approach in simple multi-object datasets. It remains an intriguing direction for future research to overcome its current limitations and to uncover the full potential of distributed object-centric representation learning approaches.

## Acknowledgements

We thank David P. Reichert and Thomas Serre for their helpful guidance for re-implementing their proposed DBM model. Additionally, we thank Emiel Hoogeboom, T. Anderson Keller, Joop Pascha and Jascha Sohl-Dickstein for their valuable feedback on the manuscript.

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

# Appendix

## A   Broader Impact

The Complex AutoEncoder is a comparatively simple object discovery approach that learns to accurately and reliably separate the phase values for different objects without supervision. On the simple multi-object datasets that we consider, it achieves highly promising results. As such, we believe it opens up a new research direction for object-centric representation learning, which investigates distributed instead of slot-based representations. We further believe that the CAE outlines a general approach for object discovery that can be extended to a wide range of domains and applications, and expect that the object-centric representations that it creates from perceptual input may prove useful to create more transparent and interpretable predictions. We do not foresee any potential negative societal impacts, but acknowledge that given the broad range of possible applications of object discovery methods, there might be potential impacts that we cannot foresee at the current time.

## B   Reproducibility Statement

To ensure the reproducibility of our experiments, we provide a detailed overview of the model architectures, hyperparameters and additional implementation details in Appendix C. Besides this, we published the code used to produce the main experimental results at `https://github.com/loeweX/ComplexAutoEncoder`.

We repeated all experiments with eight different seeds to obtain stable and reproducible results. The overall computation time for all experiments, including the baselines, corresponds to approximately 40 GPU days on a Nvidia GTX 1080Ti (not including hyperparameter search and trials throughout the research process).

## C   Experimental Details

### C.1   (Complex) AutoEncoder

Table 3 shows the architecture of the Complex AutoEncoder, as well as its real-valued counterpart. We used the default parameter initialization of PyTorch for all layers, except $f_{\text{out}}$ for which we set the initial weight $w = 1$ and the initial bias $b = 0$. Additionally, we initialize all phase-biases $b_\varphi$ with zero. We introduced the magnitude bias to create a more consistent overall formulation, which applies a bias on both the magnitude and the phase. However, this term is not strictly needed, as it is effectively canceled out by the BatchNormalization that we apply in Eq. (5). As a result, we experience that the magnitude bias does not have a noticeable impact in our experiments. After the linear layers, we apply Layer Normalization (Ba et al., 2016) instead of Batch Normalization.

We optimized all hyperparameters, except the number of training steps, on the validation set of the 3Shapes dataset and subsequently applied them for the training on all datasets. Across the board, we found that each hyperparameter setting that improved the performance of the Complex AutoEncoder also improved the performance of the real-valued autoencoder and vice versa. As a result, we use the same hyperparameters to train both models.

To create the object-wise reconstructions in Fig. 5, we first cluster the features created by the encoder $f_{\text{enc}}$ by following the procedure described in Section 3.5. Then, we mask out all values that are not part of a particular cluster with zeros. Finally, we fine-tune the decoder to reconstruct individual objects given these masked-out feature vectors for 10,000 steps using Adam with a learning rate of 5e−5. Since our network generally assigns the same phase values to the same object types, we manually match the reconstructions created by the separate feature vectors to the respective objects once and use the same assignment for the rest of the dataset.

Table 3: Autoencoding architecture used for the Complex AutoEncoder, as well as the real-valued autoencoder baseline. Additionally, the 6-layer DBM uses a setup that closely resembles the encoder $f_{\text{enc}}$ as outlined in this table.

| | Layer | Feature Dimension (H × W × C) | Kernel | Stride | Padding Input / Output | Activation Function |
|---|---|---|---|---|---|---|
| $f_{\text{enc}}$ | Conv | $16 \times 16 \times 32$ | 3 | 2 | 1 / 0 | (Complex-)ReLU |
| | Conv | $16 \times 16 \times 32$ | 3 | 1 | 1 / 0 | (Complex-)ReLU |
| | Conv | $8 \times 8 \times 64$ | 3 | 2 | 1 / 0 | (Complex-)ReLU |
| | Conv | $8 \times 8 \times 64$ | 3 | 1 | 1 / 0 | (Complex-)ReLU |
| | Conv | $4 \times 4 \times 64$ | 3 | 2 | 1 / 0 | (Complex-)ReLU |
| | Reshape | $1 \times 1 \times 1024$ | - | - | - | - |
| | Linear | $1 \times 1 \times 64$ | - | - | - | (Complex-)ReLU |
| $f_{\text{dec}}$ | Linear | $1 \times 1 \times 1024$ | - | - | - | (Complex-)ReLU |
| | Reshape | $4 \times 4 \times 64$ | - | - | - | - |
| | TransConv | $8 \times 8 \times 64$ | 3 | 2 | 1 / 1 | (Complex-)ReLU |
| | Conv | $8 \times 8 \times 64$ | 3 | 1 | 1 / 0 | (Complex-)ReLU |
| | TransConv | $16 \times 16 \times 32$ | 3 | 2 | 1 / 1 | (Complex-)ReLU |
| | Conv | $16 \times 16 \times 32$ | 3 | 1 | 1 / 0 | (Complex-)ReLU |
| | TransConv | $32 \times 32 \times 1$ | 3 | 2 | 1 / 1 | (Complex-)ReLU |

**RGB images** In its current form, the CAE cannot be applied to RGB data. This restriction is largely due to its evaluation method rather than the model itself. In fact, it is possible to simply scale the input/output dimensions to three channels and train the CAE on RGB data – the problem is that it is unclear how to evaluate the resulting phase values in a convincing way. This problem arises from the re-scaling of features with small magnitudes.

During evaluation, we re-scale features with small magnitudes to avoid unnecessary noise (Section 3.5). The closer a feature is to the origin, the less its phase will influence how it will be processed by the network, and as a result, the network has no incentive to assign anything but random phase values to these features. To evaluate how well the phases separate the objects in the scene, we need to evaluate them as independently of the magnitudes as possible. To achieve this, we normalize most features such that they will fall on the unit circle. If we did not treat features with small magnitudes separately, however, these would be projected out onto the unit circle, essentially amplifying their noisiness. Instead, we re-scale them such that they remain relatively close to the origin – ensuring little noise in the evaluation, while relying minimally on the magnitudes.

This re-scaling of features with small magnitudes may lead to a trivial separation of objects in RGB images. When following a naive approach to apply the CAE to three-channel images, such as RGB, we would output three single-channel complex numbers and interpret their respective magnitudes to represent the RGB reconstructions and the phases to represent the object identity. The problem with this approach is that for objects of specific colors, the separation becomes trivial. For example, when we have a red and a blue object, the respective reconstructed values might look something like this: For the red object, the magnitudes would take on the values $[1, 0, 0]$ and the phases could take on arbitrary values $[\varphi^R, \varphi^G, \varphi^B]$. For the blue object, the magnitudes would take on the values $[0, 0, 1]$ and the phases again could take on arbitrary values $[\varphi^R, \varphi^G, \varphi^B]$. Since we need to mask out the phases that belong to a value with a magnitude $m < 0.1$ to avoid evaluating increasingly random values, this allows for a trivial separation of the blue and red object irrespective of the phases assigned to them. As a result, applying complex-valued methods to object discovery in RGB images is not straightforward. Note, however, that the CAE already makes a step forward compared to the DBM model by Reichert & Serre (2014) by being able to process grayscale images with continuous pixel values instead of only binarized ones.

## C.2   DBM

In this section, we will first discuss the limitations of our re-implementation, then outline our final experimental setting, and finally list all other setups that we considered.

**Limitations of our Re-Implementation**   To implement the Deep Boltzmann Machine (DBM) model, we closely followed the descriptions in Reichert & Serre (2014). However, due to the lack of a publicly available codebase, some aspects of the implementation remained unclear, and we used private communication with the authors, as well as extensive experimentation to fill in the gaps. Despite the considerable effort that we put into reproducing their results – involving more fine-tuning than we performed for the proposed CAE model – our implementation seems to perform worse than the original implementation, as indicated by the visual comparison in Fig. 7. We note two things about this comparison: (1) Since Reichert & Serre (2014) report stability issues for their implementation and state that they only show results of the "well performing networks", we accordingly chose the seed with the best performance in our implementation to create the samples for the qualitative comparison. (2) We can not quantify the observed performance gap, as Reichert & Serre (2014) do not provide quantitative results in their paper.

Independent of the implementation, the DBM model is less efficient than the CAE by design, due to its greedy layerwise training with Contrastive Divergence that requires a comparatively large number of training steps, and the iterative settling procedure used for evaluation.

**Final Setup**   We implement all layers as fully-connected layers with a potentially restricted receptive field as proposed by Reichert & Serre (2014). To achieve this, we mask out all weights outside the receptive field with zero. This results in a setup that resembles a convolutional layer without shared weights. Additionally, we apply a separate bias to each spatial location. We implement these layers in two different architectures. First, we closely followed the model description provided by Reichert & Serre (2014) and use a three-layer architecture (3-layer DBM). The layers in this architecture have a receptive field size in [height, width] of $[7, 7]$, $[7, 7]$ and $[20, 20]$ (i.e. global), and hidden dimensions in [height, width, channel] or [channel] of $[26, 26, 3]$, $[20, 20, 5]$, and $[676]$, respectively. As a second architecture, we test a DBM whose layers resemble the setup of the encoder $f_{enc}$ of the (Complex) AutoEncoder (6-layer DBM, Table 3). Instead of convolutional layers, we use the fully-connected layers with restricted receptive field sizes as described above, where the kernel size corresponds to the receptive field size. Note that even though this setup resembles that of the CAE, its parameter count differs considerably as weights are not shared across locations, but they are shared for the forward and backward pass (i.e. encoding and decoding). For all layers in both models, we use a sigmoid non-linearity as activation function.

We train each layer greedily as a separate Restricted Boltzmann Machine (RBM) using 1-step Contrastive Divergence (Hinton, 2012) with a learning rate of $1e-1$, momentum coefficient of 0.5 and weight-decay factor of $1e-4$, for 50,000 and 100,000 steps. After training, to create object-centric representations, we inject complex-valued activations into the model. To do so, we use the input images as magnitudes, randomly sample phase values for each pixel from $\mathcal{U}(0, 2\pi)$ and initialize the hidden state of each layer with an initial forward pass. Then, clamping the magnitudes of the visible units to the input images, we iterate the complex-valued activations 500 times through the model. To evaluate the resulting phase values, we use the same discretization procedure as for the CAE (see Section 3.5). This procedure includes a step in which small magnitudes are used to mask out their corresponding phase values. Since we use the final output of the DBM to extract both the phases and magnitudes, and not some intermediate values as in the CAE, this model creates a good foreground-background separation by relying explicitly on a black background – resulting in small magnitudes for the reconstruction and thus masked out phases. Thus, the ARI+BG performance of the DBM models is negligible, as it depends directly on its reconstruction performance rather than its object discovery performance.

Finally, since the DBM is only well-defined on binary inputs, we apply this model to a binarized version of the MNIST&Shape dataset. To create this version of the dataset, we use the same threshold as for the creation of the pixel-wise labels (-0.8 after normalization to the [-1,1] range, see Appendix C.4) to decide whether a pixel is assigned a "0" or "1" value.

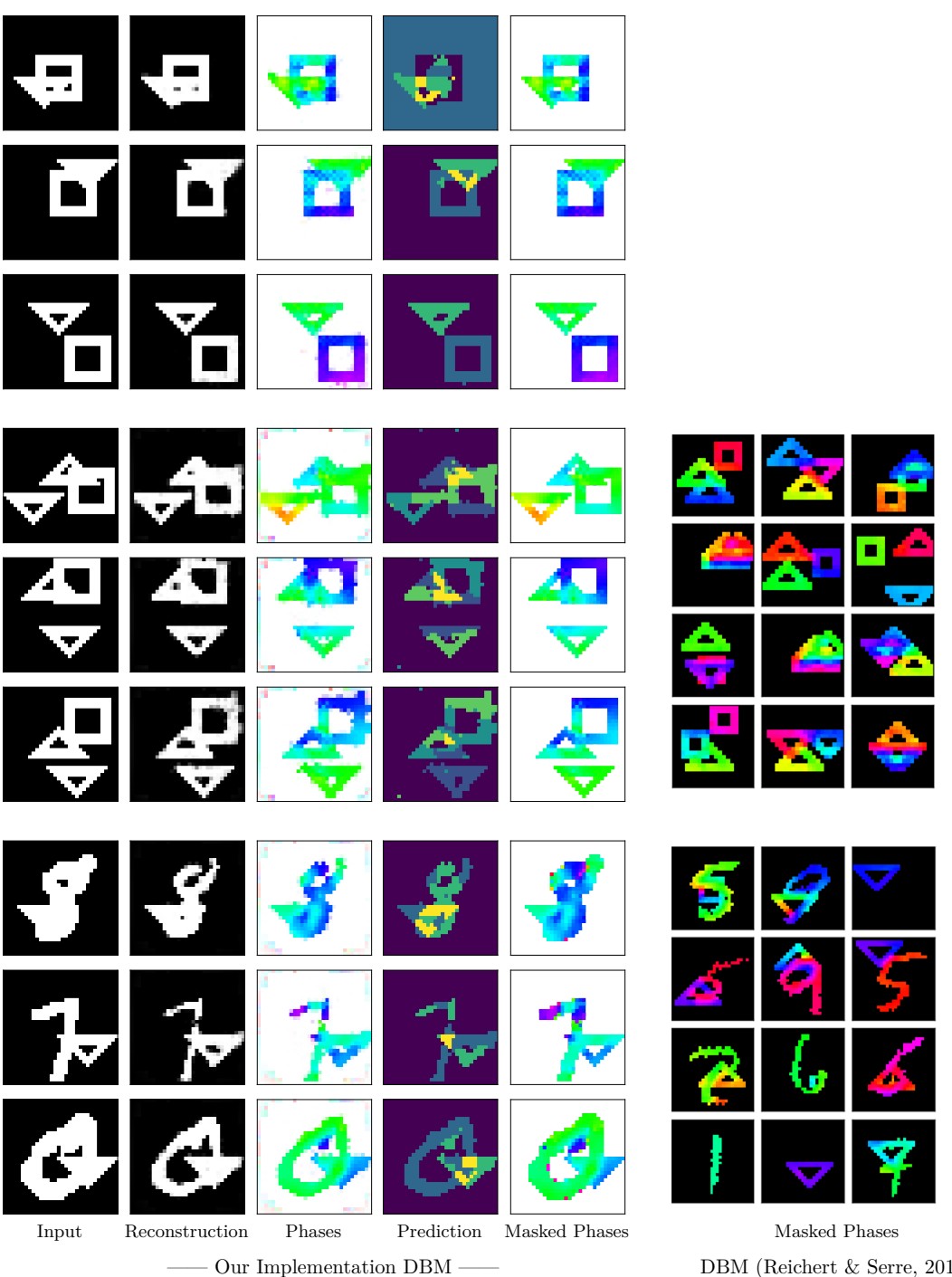

Input  Reconstruction  Phases  Prediction  Masked Phases        Masked Phases

—— Our Implementation DBM ——                DBM (Reichert & Serre, 2014)

Figure 7: Qualitative comparison between random samples from the best performing seed of our re-implementation of the DBM model in Reichert & Serre (2014) (**left**) and original samples taken from their paper on slightly different datasets (**right**). To create the "Masked Phases" images, we follow Reichert & Serre (2014) and use the input values (and thus the fact that objects are white, and the background is black) to mask out the phase values of the background pixels. Despite achieving a reasonable reconstruction performance on all three tested datasets, the object discovery performance of our re-implementation seems to lag behind the original implementation.

**Tested Setups** Besides this final setup which achieved the best validation performance, we have tried a wide range of other experimental settings:

- We trained the 3-layer DBM model for up to 1,500,000 steps. Interestingly, even though the reconstruction performance keeps on improving, object discovery performance worsens (similarly to the effects observed in Table 6).

- We tested different learning rates $[1, 1e{-}1, 1e{-}2, 1e{-}3, 1e{-}4]$ and different momentum terms $[0.5, 1]$.

- We implemented and tested 5-step persistent contrastive divergence (Tieleman, 2008), an alternative training algorithm for RBMs which was shown to achieve better results, with different learning rates, momentum terms and learning rate schedules (fixed and exponential decay).

- We tried to share the weights and biases across spatial locations to create layers that resemble a convolutional layer more closely.

- We tested a different initialization of the hidden states during evaluation, in which we randomly sampled magnitudes from a Bernoulli distribution with $p = 0.5$ and phase values from $\mathcal{U}(0, 2\pi)$.

- During evaluation, we ran the model for different numbers of iterations (100, 500, 1000), and chose the number that gave the best trade-off between performance and speed.

### C.3 SlotAttention

To implement SlotAttention, we followed the description and hyperparameters provided by Locatello et al. (2020) as well as their open-source implementation[3]. We used a hidden dimension of 64 throughout the model and adjusted the decoding architecture as described in Table 4 as this improved SlotAttention's performance on our datasets. Besides this final setup that we found to perform best in terms of object discovery performance, we tested the following setups on the validation set of the 3Shapes dataset: the decoder setup as used by Locatello et al. (2020) for the Tetrominoes and Multi-dSprites datasets (i.e. spatially broadcast to a resolution of $32 \times 32$ and apply four transposed-convolutional layers), a setup in which the fourth transposed-convolutional layer in Table 4 is removed from the decoder, as well as a setup in which the number of channels is halved across all layers. None of these setups learned to disentangle objects on the MNIST&Shape dataset. Note that, since we select the SlotAttention architecture based on its object discovery performance, its reconstruction performance is not comparable to the CAE due to a different number of learnable parameters and a different bottleneck size.

Table 4: Architecture used for the Spatial-Broadcast Decoder in the SlotAttention model.

| Layer | Feature Dimension (H × W × C) | Kernel | Stride | Padding Input / Output | Activation Function |
|---|---|---|---|---|---|
| Spatial Broadcast | 4 × 4 × 64 | - | - | - | - |
| Position Embedding | 4 × 4 × 64 | - | - | - | |
| TransConv | 7 × 7 × 64 | 5 | 2 | 2 / 0 | ReLU |
| TransConv | 15 × 15 × 64 | 5 | 2 | 2 / 0 | ReLU |
| TransConv | 32 × 32 × 64 | 5 | 2 | 2 / 1 | ReLU |
| TransConv | 32 × 32 × 64 | 3 | 1 | 1 / 0 | ReLU |
| TransConv | 32 × 32 × 2 | 3 | 1 | 1 / 0 | ReLU |

**Investigating the failure on the MNIST&Shape dataset** To investigate the different factors that lead to SlotAttention's failure on the MNIST&Shape dataset, we ran four additional experiments (Table 5). First, we trained SlotAttention on a version of the MNIST&Shape dataset in which the MNIST digits are downsized to $16 \times 16$ pixels (*small MNIST&Shape*). On this dataset, SlotAttention fails to separate the two objects. Second, we trained SlotAttention on a version of the MNIST&Shape dataset in which the MNIST

---

[3]https://github.com/google-research/google-research/tree/master/slot_attention

Table 5: Investigating the failure of SlotAttention on the MNIST&Shape dataset. The results on the 2Shapes-randBG dataset and across variations of the MNIST&Shape dataset (mean $\pm$ sem across 8 seeds) indicate that SlotAttention struggles with large objects and grayscale images.

| Dataset | MSE $\downarrow$ | ARI+BG $\uparrow$ | ARI-BG $\uparrow$ |
|---|---|---|---|
| small MNIST&Shape | $4.495\text{e-}03 \pm 4.804\text{e-}04$ | $0.203 \pm 0.086$ | $0.365 \pm 0.088$ |
| MNIST&Shape binarized | $3.071\text{e-}02 \pm 1.126\text{e-}03$ | $0.081 \pm 0.018$ | $0.153 \pm 0.038$ |
| small MNIST&Shape binarized | $1.025\text{e-}02 \pm 7.853\text{e-}04$ | $0.380 \pm 0.108$ | $0.851 \pm 0.022$ |
| 2Shapes randBG | $1.190\text{e-}05 \pm 2.235\text{e-}06$ | $0.538 \pm 0.174$ | $0.604 \pm 0.166$ |
| 4 working seeds: 2Shapes randBG | $1.526\text{e-}05 \pm 3.598\text{e-}06$ | $0.997 \pm 0.001$ | $1.000 \pm 0.000$ |
| 4 failing seeds: 2Shapes randBG | $8.539\text{e-}06 \pm 1.685\text{e-}06$ | $0.078 \pm 0.032$ | $0.208 \pm 0.154$ |

digits are of the original size, but binarized (*MNIST&Shape binarized*). Again, SlotAttention fails on this dataset. If we combine the two modifications and create a dataset with smaller, binarized MNIST digits, SlotAttention finally starts to separate the objects (*small MNIST&Shape binarized*). However, it still does not perform perfectly – in some cases, it splits the digits into two objects, e.g. by splitting an eight into two separate circles. Last but not least, we applied SlotAttention on the *2Shapes-randBG* dataset. Here, SlotAttention performs perfectly for half of the seeds; and largely fails for the other half. Interestingly, the seeds that fail in terms of object discovery performance achieve a better reconstruction performance.

Based on these observations, we believe that SlotAttention fails on the MNIST&Shape dataset due to two factors: (1) the MNIST digits are too large to be covered by SlotAttention's receptive field. Only when the MNIST digits are downsized, can the SlotAttention model learn to separate the objects; (2) the SlotAttention model performs poorly on grayscale values, as highlighted by its results on the non-binarized MNIST&Shape datasets and the 2Shapes-randBG dataset. We hypothesize that this is due to the alpha mask that is used in the spatial-broadcast decoder: the SlotAttention model might learn to use this mask to reconstruct the precise grayscale value rather than to correctly merge the reconstructions from the individual slots. This problem would be naturally circumvented by RGB data.

### C.4 Datasets

For our experiments, we generate three grayscale datasets: 2Shapes, 3Shapes, and MNIST&Shape. All images within these datasets feature a black background and white objects of differing shapes. In the 2Shapes and 3Shapes datasets, the foreground objects and the background are plain white and plain black, respectively, without noise. In the MNIST&Shape dataset, the digits exhibit differing grayscale values. All objects are placed in random locations while ensuring that no part of the object is cut-off at the image boundary.

We use four different object types ($\square, \triangle, \triangledown$, and MNIST digits). The square has an outer side-length of 13 pixels. Both triangles are isosceles triangles, have a base-length of 17 pixels, and are 9 pixels high. Both the square's and the triangles' outlines have a width of 3 pixels.

For the MNIST&Shape dataset, we resize each MNIST digit to match the input image size of our dataset (i.e. $32 \times 32$) before applying it to an image. Then, we label pixels as "digit" when their value is $> -0.8$ after normalization to the $[-1, 1]$ range. This threshold ensures that most of the digit pixels are labeled as such, while minimizing the influence of potentially noisy background pixels. We follow the original dataset split to create the test images and divide the original training set to get 50,000 MNIST digits for our training set and 10,000 MNIST digits for our validation set.

We scale and shift all inputs to the range $[0, 1]$ for the autoencoding models, and we use an input range of $[-1, 1]$ for the SlotAttention model.

# D  Additional Results

## D.1  Global Rotation Equivariance

In this section, we will first proof that the CAE model is equivariant to global rotations, before verifying this property empirically.

### D.1.1  Proof of Global Rotation Equivariance

Before giving the proof for Proposition 3.1, we formally define global rotation equivariance.

**Definition D.1.** (Global Rotation Equivariance) A function $f : \mathbb{C}^a \to \mathbb{C}^b$ is equivariant w.r.t. global rotations if for any arbitrary phase offset $\alpha \in [0, 2\pi)$ it holds that:

$$f(\mathbf{x} \cdot (\cos(\alpha) + \sin(\alpha)i)) = f(\mathbf{x}) \cdot (\cos(\alpha) + \sin(\alpha)i) \tag{7}$$

*Proof.* To proof that the CAE autoencoder is equivariant w.r.t. global rotations, we consider each operation performed in its layers separately. If all of these operations are equivariant, their composite function (i.e. the overall model) is equivariant as well.

Equation (2) describes the way that the weights are applied to the input. We consider the simple setting of a fully connected layer, which extends trivially to other layer types such as convolutions and transposed convolutions:

$$f_{\mathbf{w}}(\mathbf{z}) \cdot (\cos(\alpha) + \sin(\alpha)i) = \langle \mathbf{w}, \mathbf{z} \rangle \cdot (\cos(\alpha) + \sin(\alpha)i) \tag{8}$$

Due to the complex multiplication being commutative, we get:

$$= \langle \mathbf{w}, \mathbf{z} \cdot (\cos(\alpha) + \sin(\alpha)i) \rangle \tag{9}$$
$$= f_{\mathbf{w}}(\mathbf{z} \cdot (\cos(\alpha) + \sin(\alpha)i)) \tag{10}$$

Equation (3) applies the bias terms to $\boldsymbol{\psi}$:

$$\boldsymbol{m}_{\psi} = |\boldsymbol{\psi}| + \boldsymbol{b_m} \quad \in \mathbb{R}^{d_{\text{out}}} \qquad \boldsymbol{\varphi}_{\psi} = \arg(\boldsymbol{\psi}) + \boldsymbol{b_\varphi} \quad \in \mathbb{R}^{d_{\text{out}}} \tag{11}$$

The addition of the magnitude bias $\boldsymbol{b_m}$ is equivariant w.r.t. global rotations, since the rotation by any angle leaves the magnitude unchanged. The addition of the phase bias $\boldsymbol{b_\varphi}$ is rotation equivariant, as it applies a fixed shift to the phases independent of their values.

Equation (4) describes the application of the network to the magnitudes of the input features:

$$\boldsymbol{\chi} = f_{\mathbf{w}}(|\mathbf{z}|) + \boldsymbol{b_m} \qquad\qquad \in \mathbb{R}^{d_{\text{out}}} \tag{12}$$
$$\boldsymbol{m_z} = 0.5 \cdot \boldsymbol{m}_{\psi} + 0.5 \cdot \boldsymbol{\chi} \qquad\qquad \in \mathbb{R}^{d_{\text{out}}} \tag{13}$$

Similarly to above, this computation is equivariant w.r.t. global rotations, since the rotation by any angle leaves the magnitudes unchanged.

Equation (5) applies a non-linearity on the activations:

$$\mathbf{z}' = \text{ReLU}(\text{BatchNorm}(\boldsymbol{m_z})) \circ e^{i\boldsymbol{\varphi}_\psi} \qquad\qquad \in \mathbb{C}^{d_{\text{out}}} \tag{14}$$

Since this non-linearity only affects the magnitudes, but leaves the phase values unchanged, this function is equivariant to global rotations (i.e. phase shifts) as well.

Thus, all operations within the CAE are equivariant w.r.t. global rotations, and as a result, the CAE model is equivariant w.r.t. global rotations. $\qquad\square$

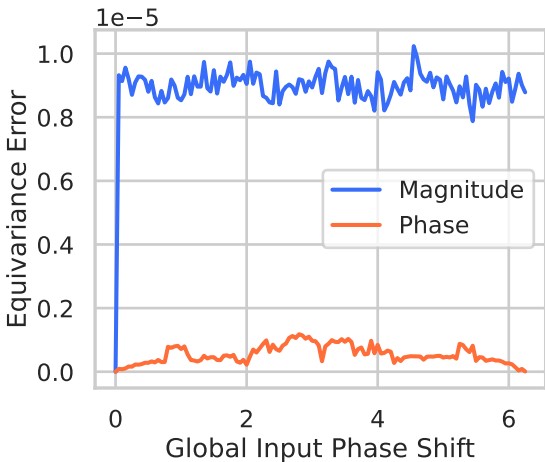

Figure 8: Equivariance error of the CAE. We measure the average $L^2$-distance (i.e. equivariance error, y-axis) between the output magnitudes and phases created with a default input phase value of zero and outputs created with input phase shifts (x-axis) between 0 and $2\pi$. We observe that the equivariance error is minimal for both the magnitude and phase.

### D.1.2 Empirical Validation of Global Rotation Equivariance

We verify empirically that the CAE is equivariant w.r.t. global rotations. For this, we take a randomly initialized CAE model and input batches of random noise images with various phase values. Then, we measure the average magnitude and phase difference between the default input phase of zero and input phase interpolations between zero and $2\pi$. More precisely, we measure the $L^2$-distance between the individual pixel magnitudes and phases, and aggregate the results into an average error per global phase shift. For the equivariance error of the phase, we further subtract the expected shift from the output phases. As can be seen from Fig. 8, the resulting equivariance error for both the magnitude and phase are minimal.

### D.2 DBM

We test two architectures for the DBM model, 3-layer DBM and 6-layer DBM, trained for 50,000 and 100,000 steps per layer (Table 6). On the 2Shapes dataset, the 3-layer DBM trained for 100,000 steps per layer (i.e. 300,000 steps in total) achieves the best reconstruction (MSE) and object discovery (ARI-BG) performance. On the 3Shapes and MNIST&Shape datasets, the 6-layer DBM achieves the best object discovery results in terms of ARI-BG, despite creating less accurate reconstructions. In fact, for the 6-layer DBM model trained on the 3Shapes dataset, the object discovery performance in terms of ARI-BG worsens when the model is trained for longer, despite an improvement in reconstruction performance. In the main results in Table 1, we state the performance of the best model in terms of ARI-BG performance per dataset.

Table 6: MSE and ARI scores (mean ± standard error across 8 seeds) of the two DBM models (3-layer DBM and 6-layer DBM) across the three tested multi-object datasets. On the 2Shapes dataset, the 3-layer DBM achieves the best reconstruction and object discovery performance in terms of ARI-BG, whereas the 6-layer DBM achieves better object discovery performance on the 3Shapes and MNIST&Shape dataset. Interestingly, training the 6-layer DBM model for longer on the 3Shapes dataset results in a worse object discovery performance in terms of ARI-BG despite improvements in the reconstruction performance.

| Dataset | Model | Steps | MSE ↓ | ARI+BG ↑ | ARI-BG ↑ |
|---|---|---|---|---|---|
| 2Shapes | Complex AutoEncoder | 10000 | **3.322e-04** ± 1.583e-06 | **0.999** ± 0.000 | **1.000** ± 0.000 |
| | 3-layer DBM | 150000 | 4.358e-03 ± 1.068e-04 | 0.893 ± 0.002 | 0.694 ± 0.017 |
| | 3-layer DBM | 300000 | 3.308e-03 ± 1.024e-04 | 0.920 ± 0.002 | 0.744 ± 0.010 |
| | 6-layer DBM | 300000 | 8.699e-03 ± 1.442e-04 | 0.893 ± 0.001 | 0.689 ± 0.015 |
| | 6-layer DBM | 600000 | 4.892e-03 ± 5.394e-05 | 0.934 ± 0.001 | 0.696 ± 0.013 |
| 3Shapes | Complex AutoEncoder | 100000 | **1.313e-04** ± 2.020e-05 | **0.976** ± 0.002 | **1.000** ± 0.000 |
| | 3-layer DBM | 150000 | 6.644e-03 ± 1.618e-04 | 0.866 ± 0.001 | 0.333 ± 0.016 |
| | 3-layer DBM | 300000 | 6.111e-03 ± 2.530e-04 | 0.888 ± 0.002 | 0.377 ± 0.007 |
| | 6-layer DBM | 300000 | 1.045e-02 ± 1.494e-04 | 0.856 ± 0.006 | 0.419 ± 0.023 |
| | 6-layer DBM | 600000 | 6.206e-03 ± 1.058e-04 | 0.913 ± 0.001 | 0.397 ± 0.009 |
| MNIST&Shape | Complex AutoEncoder | 10000 | **3.185e-03** ± 1.514e-04 | **0.783** ± 0.004 | **0.971** ± 0.011 |
| | 3-layer DBM | 150000 | 1.381e-02 ± 6.307e-05 | 0.762 ± 0.002 | 0.047 ± 0.005 |
| | 3-layer DBM | 300000 | 1.321e-02 ± 6.240e-05 | 0.755 ± 0.001 | 0.065 ± 0.008 |
| | 6-layer DBM | 300000 | 1.839e-02 ± 1.171e-04 | 0.697 ± 0.003 | 0.153 ± 0.010 |
| | 6-layer DBM | 600000 | 1.560e-02 ± 8.069e-05 | 0.718 ± 0.002 | 0.175 ± 0.006 |

## D.3 Model Sensitivity Analysis

Fig. 9 highlights the influence of the feature dimension that is output by the CAE's encoder $f_{enc}$ on the model's performance. We find that the model achieves strong performance for a broad range of feature dimensions – indicating that the CAE does not require a restricted bottleneck size to create disentangled object representations.

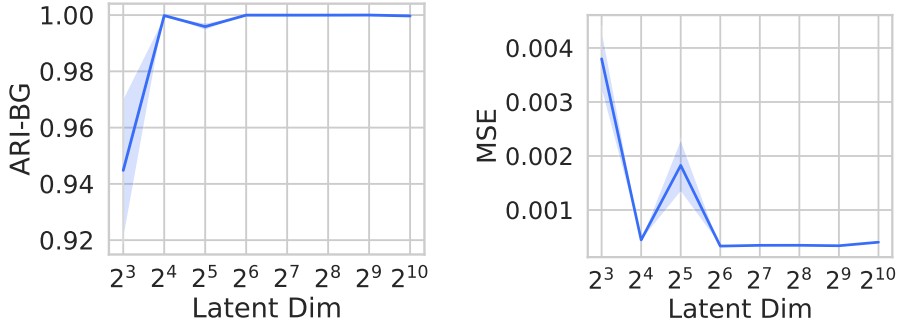

Figure 9: Influence of the latent dimension on CAE's performance on the 2Shapes dataset (mean ± standard error across 8 seeds). We find that the model does not require a restricted bottleneck to create disentangled object representations.

## D.4 Dataset Sensitivity Analysis

To explore the capabilities and limitations of the Complex AutoEncoder under various conditions, we apply it to a set of new datasets. In Table 7, we provide a detailed breakdown of all results and in Figs. 10 and 11, we show qualitative results for each dataset.

Below, we provide additional details for each dataset:

For the **2Shapes-randBG** dataset, we vary the 2Shapes dataset by applying a different background color to each image. This background color is randomly sampled from the $[0, 0.75)$ range – given that the pixel

Table 7: Results on the exploratory datasets investigating capabilities and limitations of the CAE (mean $\pm$ standard error across 8 seeds). We find that the CAE can handle a wide variety of dataset settings, such as variable backgrounds, repeated objects and decreasing objects counts. However, it struggles to represent a larger number of objects at a time and to generalize to more objects than observed during training.

| Dataset | MSE $\downarrow$ | ARI+BG $\uparrow$ | ARI-BG $\uparrow$ |
|---|---|---|---|
| 2Shapes-randBG | $4.805\mathrm{e}{-04}_{\ \pm\ 6.390\mathrm{e}{-05}}$ | $0.971_{\ \pm\ 0.002}$ | $0.995_{\ \pm\ 0.001}$ |
| 2Triangles | $4.441\mathrm{e}{-06}_{\ \pm\ 4.046\mathrm{e}{-06}}$ | $0.879_{\ \pm\ 0.019}$ | $0.948_{\ \pm\ 0.045}$ |
| 4Shapes | $3.242\mathrm{e}{-03}_{\ \pm\ 1.324\mathrm{e}{-04}}$ | $0.860_{\ \pm\ 0.014}$ | $0.669_{\ \pm\ 0.029}$ |
| 2/5Shapes | $2.463\mathrm{e}{-04}_{\ \pm\ 4.767\mathrm{e}{-05}}$ | $0.939_{\ \pm\ 0.014}$ | $0.969_{\ \pm\ 0.007}$ |
| >2Shapes | $3.979\mathrm{e}{-02}_{\ \pm\ 9.438\mathrm{e}{-04}}$ | $0.844_{\ \pm\ 0.008}$ | $0.758_{\ \pm\ 0.007}$ |
| $\leq$3Shapes Test | $4.702\mathrm{e}{-03}_{\ \pm\ 8.866\mathrm{e}{-04}}$ | $0.923_{\ \pm\ 0.011}$ | $0.956_{\ \pm\ 0.019}$ |

values of the images range from 0 to 1 and that objects have a color value of 1, this ensures that background colors vary maximally while the objects stay clearly discernible from the background. After training on this dataset for 10,000 steps, the CAE creates accurate reconstructions and clearly separates the objects through its assigned phase values, highlighting that it can handle variable backgrounds. Interestingly, for brighter background colors, the model also learns to assign a separate phase to the background – enabling the model to differentiate foreground from background even in this more challenging setting.

For the **2Triangles** dataset, we place two downward-facing triangles in each image and train the model for 100,000 steps. Given the CAE's high performance on this dataset, we conclude that it can separately represent multiple instances of the same object co-occuring in the same image.

The **4Shapes** dataset builds on the 3Shapes dataset and adds an additional circle with an outer radius of 6 pixels to each image. Similarly to the 3Shapes dataset, we train the model for 100,000 steps. On this dataset, the performance of the CAE drops significantly – indicating that it cannot represent this amount of objects at a time.

For the **2/5Shapes** dataset, we randomly sample two shapes for each image out of a set of five possible shapes, the same four shapes as in the 4Shapes dataset plus an additional larger circle with an outer radius of 10 pixels. After 100,000 training steps, the CAE achieves a good performance on this dataset indicating that it can learn to represent a larger set of objects as long as the number of objects per image is limited.

To test the generalization capabilities of the CAE, we create the >2Shapes and $\leq$3Shapes datasets. For the **>2Shapes** dataset, we train the CAE on the 2Shapes dataset and test it on a dataset in which the downward-facing triangle that appears in the 2Shapes dataset appears twice in each image (and thus we have three objets per image). The relatively low performace of the CAE on this dataset indicates that it cannot generalize well to more objects that observed during trainig.

For the **$\leq$3Shapes** dataset, we use the same set of objects as in the 3Shapes dataset, but randomly decide whether to place one, two or three of these objects into each image. We evaluate the CAE's generalization performance on this dataset by training it on the 3Shapes dataset for 100,000 steps, and subsequently testing it on the $\leq$3Shapes dataset. Based on the strong performance of the CAE, we conclude that it can handle variable numbers of objects in images and that it can generalize seamlessly to a setting in which fewer objects are present than observed during training.

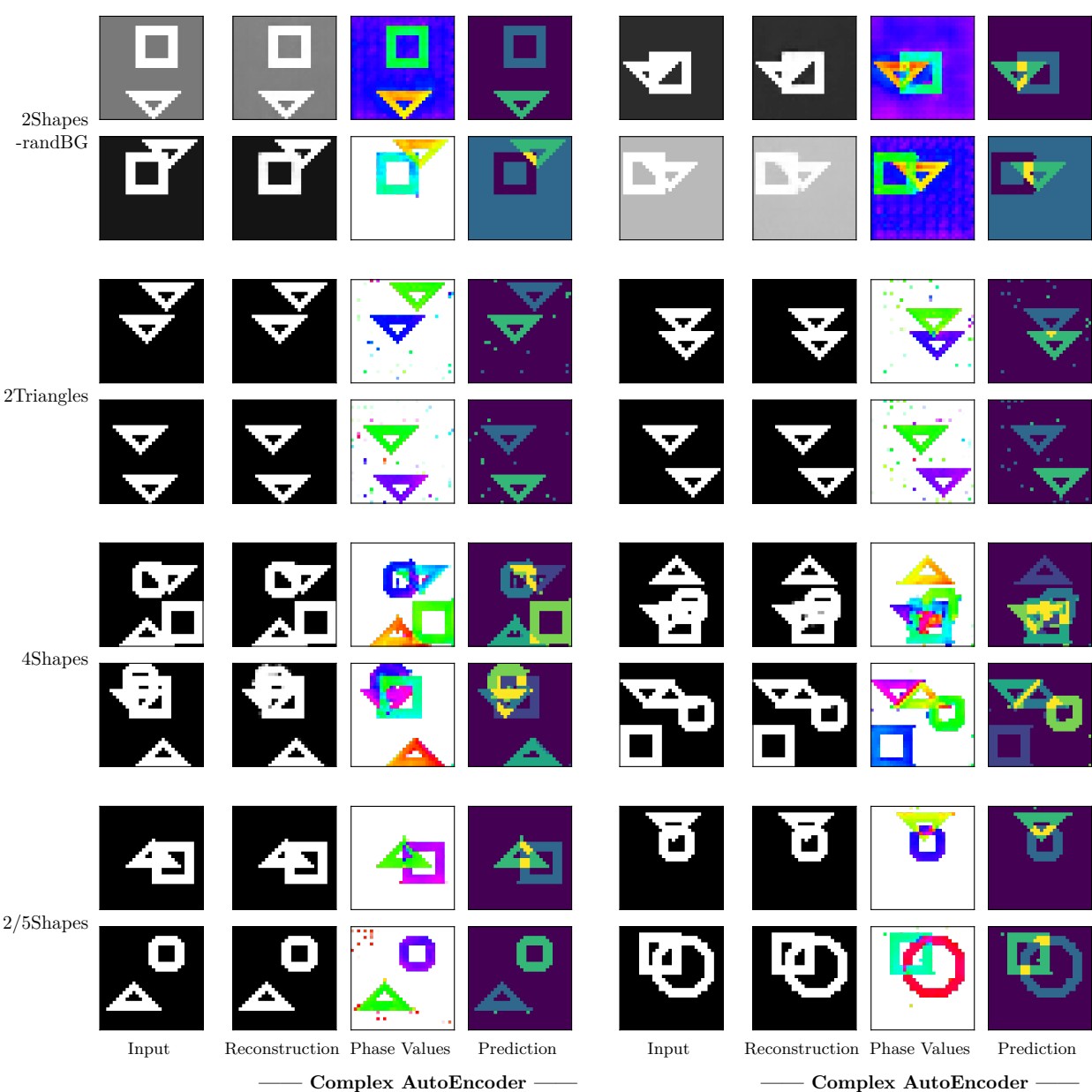

Figure 10: Qualitative results for the dataset sensitivity analysis of the Complex AutoEncoder. The depicted images are test-samples which were hand-selected to best highlight the variability of samples within each dataset.

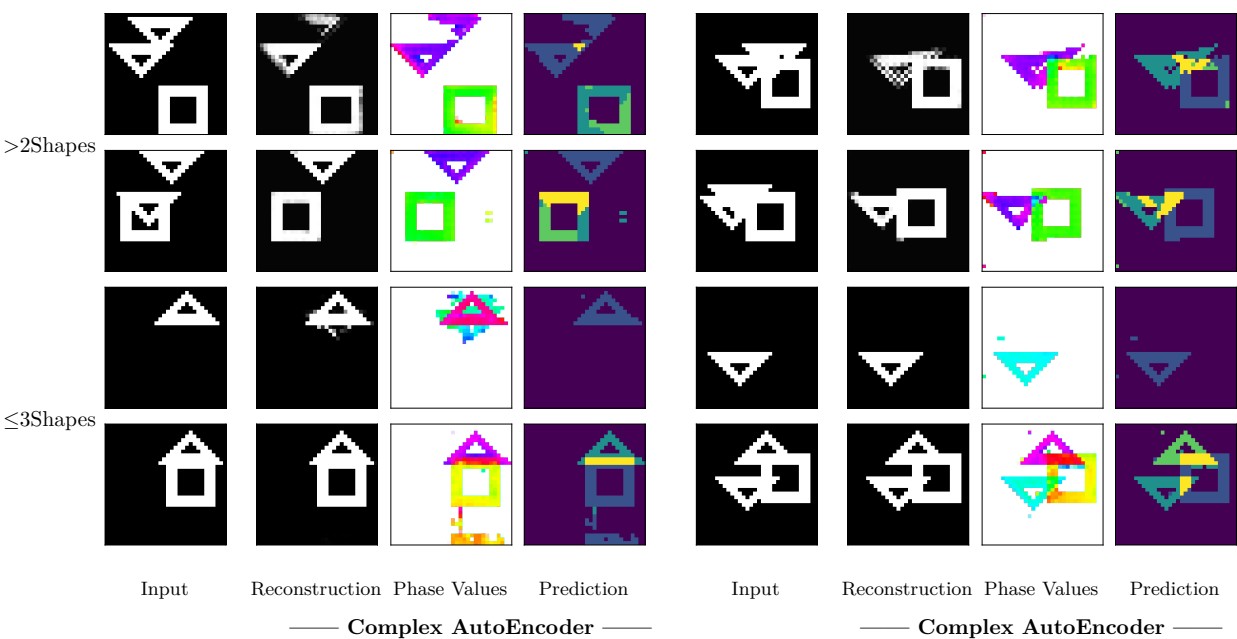

Figure 11: Qualitative results for the dataset sensitivity analysis of the Complex AutoEncoder. The depicted images are test-samples which were hand-selected to best highlight the variability of samples within each dataset. While the CAE does not generalize to a dataset depicting more objects (**>2Shapes**), it does generalize to a smaller object count than observed during training (**≤3Shapes**).

### D.5 Runtimes

To compare the training times between models, we first investigate the number of training steps that they require to achieve their best performance. For this, we plot the training curves of the Complex AutoEncoder and SlotAttention model on the 2Shapes and 3Shapes datasets in Fig. 12. In both datasets, the SlotAttention model keeps improving in performance throughout its 500,000 training steps. The Complex AutoEncoder, on the other hand, converges much faster, within 10,000 – 100,000 steps. For the DBM model, we cannot provide training curves due to its greedy layerwise training. Instead, we evaluate its performance after 50,000 and 100,000 training steps per layer in Table 6 and find that longer training times generally result in improved reconstruction performance. For the final run-time comparison, we choose the DBM model with the best ARI-BG performance per dataset.

We provide a comparison of the training times of the Complex AutoEncoder, SlotAttention and DBM model in Table 8. We find that the DBM models have the fastest training time per 10,000 steps, but are overall slower to train than the CAE due to the high number of training steps that they require. SlotAttention's training steps are two times slower than those of the CAE, which, together with its high number of training steps, leads to significantly slower training times.

In Table 9, we compare the evaluation times of each model on the 2Shapes dataset. Since all datasets in our experiments use the same dimensionality ($32 \times 32$ pixels) and contain 10,000 images, we expect these numbers to generalize well across the tested datasets. Since SlotAttention creates an explicit separation of object features into slots, its predicted object assignments are readily available and a single evaluation of the test set takes approximately 25.4 seconds. Both the CAE and DBM model, on the other hand, need to extract discrete object assignments from the continuous phase values using $k$-means, resulting in considerably longer evaluation times. While the CAE takes approximately 263.9 seconds to evaluate, due to its iterative settling procedure, the DBM takes between 520.9 and 4377.6 seconds – depending on whether we use the 3-layer or 6-layer variant.

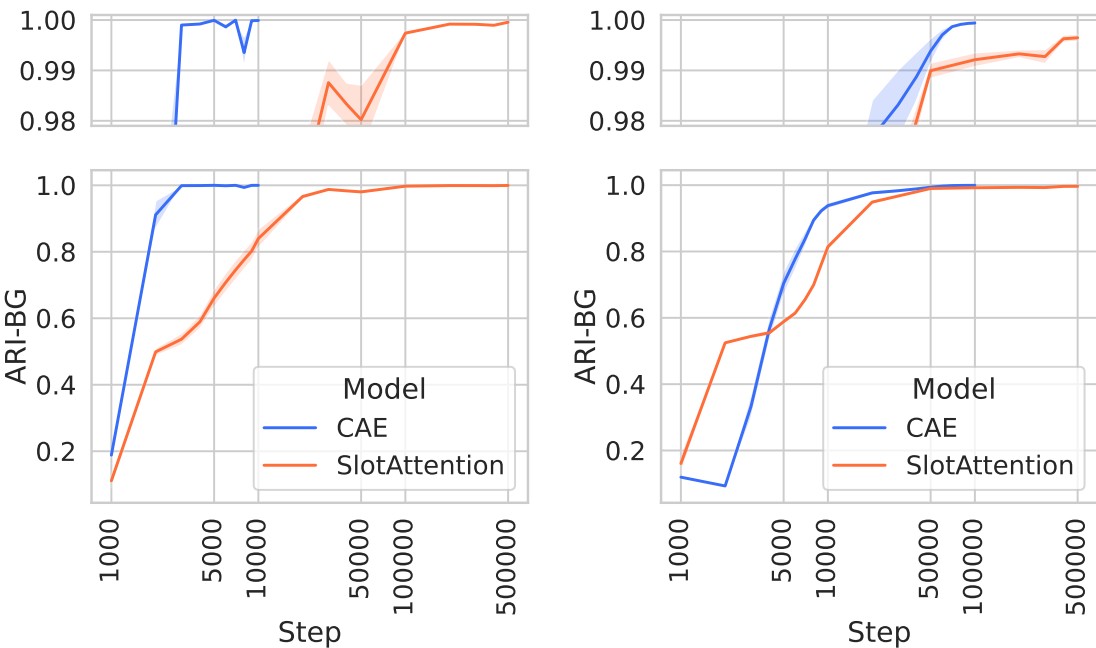

Figure 12: Training curves of the Complex AutoEncoder (CAE) and SlotAttention model (mean ± standard error across 8 seeds) on the 2Shapes (**Left**) and 3Shapes (**Right**) dataset. We plot the ARI-BG scores on the validation set throughout training. The CAE achieves comparable or better performance within 5-50 times fewer training steps compared to the SlotAttention model.

Table 8: Training times (mean ± standard error across 8 seeds) in terms of wall clock time for the Complex AutoEncoder, SlotAttention and DBM model across the three tested multi-object datasets. We measured the training times per 10,000 steps on the 2Shapes dataset and extrapolated the results to create a detailed and realistic break-down of training times for each dataset. This is possible, as all datasets are of the same dimensionality. For the DBM, we report the average training time per step across layers. We find that the CAE is considerably faster to train on all datasets compared to the SlotAttention and DBM model.

| Dataset | Model | Steps | Training Time per 10,000 Steps (sec) | Total Training Time (min) | Relative Training Speed compared to CAE |
|---|---|---|---|---|---|
| 2Shapes | Complex AutoEncoder | 10000 | $456.6_{\pm 2.5}$ | $7.6_{\pm 0.0}$ | $1.0_{\pm 0.0}$ |
| | 3-layer DBM | 300000 | $31.5_{\pm 0.2}$ | $15.7_{\pm 0.1}$ | $2.1_{\pm 0.0}$ |
| | SlotAttention | 500000 | $911.1_{\pm 4.5}$ | $759.3_{\pm 3.7}$ | $99.8_{\pm 0.5}$ |
| 3Shapes | Complex AutoEncoder | 100000 | $456.6_{\pm 2.5}$ | $76.1_{\pm 0.4}$ | $1.0_{\pm 0.0}$ |
| | 6-layer DBM | 300000 | $160.0_{\pm 0.3}$ | $80.0_{\pm 0.1}$ | $1.1_{\pm 0.0}$ |
| | SlotAttention | 500000 | $911.1_{\pm 4.5}$ | $759.3_{\pm 3.7}$ | $10.0_{\pm 0.0}$ |
| MNIST&Shape | Complex AutoEncoder | 10000 | $456.6_{\pm 2.5}$ | $7.6_{\pm 0.0}$ | $1.0_{\pm 0.0}$ |
| | 6-layer DBM | 600000 | $160.0_{\pm 0.3}$ | $160.0_{\pm 0.3}$ | $21.0_{\pm 0.0}$ |
| | SlotAttention | 500000 | $911.1_{\pm 4.5}$ | $759.3_{\pm 3.7}$ | $99.8_{\pm 0.5}$ |

### D.6 Additional Qualitative Results

We highlight the phase separations created by the Complex AutoEncoder in Fig. 13, object-wise reconstructions in Fig. 14, and compare all models on the 2Shapes, 3Shapes and MNIST&Shape datasets in Figures 15, 16 and 17, respectively.

Table 9: Evaluation times (mean ± standard error across 8 seeds) in terms of wall clock time for the Complex AutoEncoder, SlotAttention and DBM model on the 2Shapes dataset. Since all datasets are of the same dimensionality, we expect the evaluation times on the 3Shapes and MNIST&Shape datasets to be equivalent. The SlotAttention model is the fastest to evaluate, as it immediately creates discrete object assignments. The CAE and DBM, on the other hand, require a discretization procedure with $k$-means leading to slower evaluation times. The DBM is further slowed down by its iterative settling procedure.

| Model | Total Testing Time (sec) | Relative Testing Speed compared to CAE |
|---|---|---|
| Complex AutoEncoder | $263.9_{\pm 0.5}$ | $1.0_{\pm 0.0}$ |
| 3-layer DBM | $520.9_{\pm 1.4}$ | $2.0_{\pm 0.0}$ |
| 6-layer DBM | $4377.6_{\pm 4.5}$ | $16.6_{\pm 0.0}$ |
| SlotAttention | $25.4_{\pm 0.2}$ | $0.1_{\pm 0.0}$ |

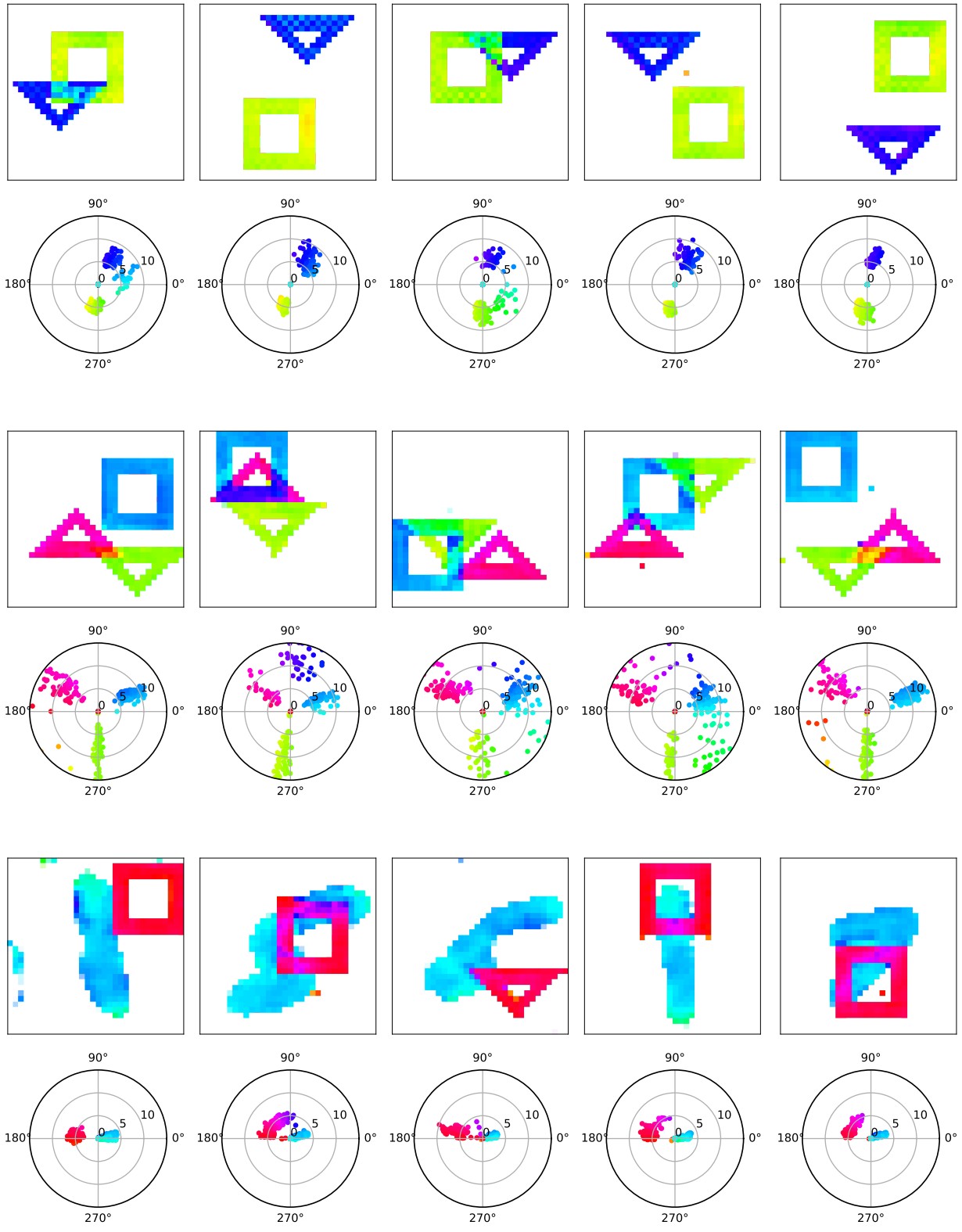

Figure 13: Phase separation in the Complex AutoEncoder on random test-samples from the 2Shapes (**Top**), 3Shapes (**Middle**) and MNIST&Shape datasets (**Bottom**). For each sample, we show the output phase images on top, and the corresponding output values in the complex plane in the bottom.

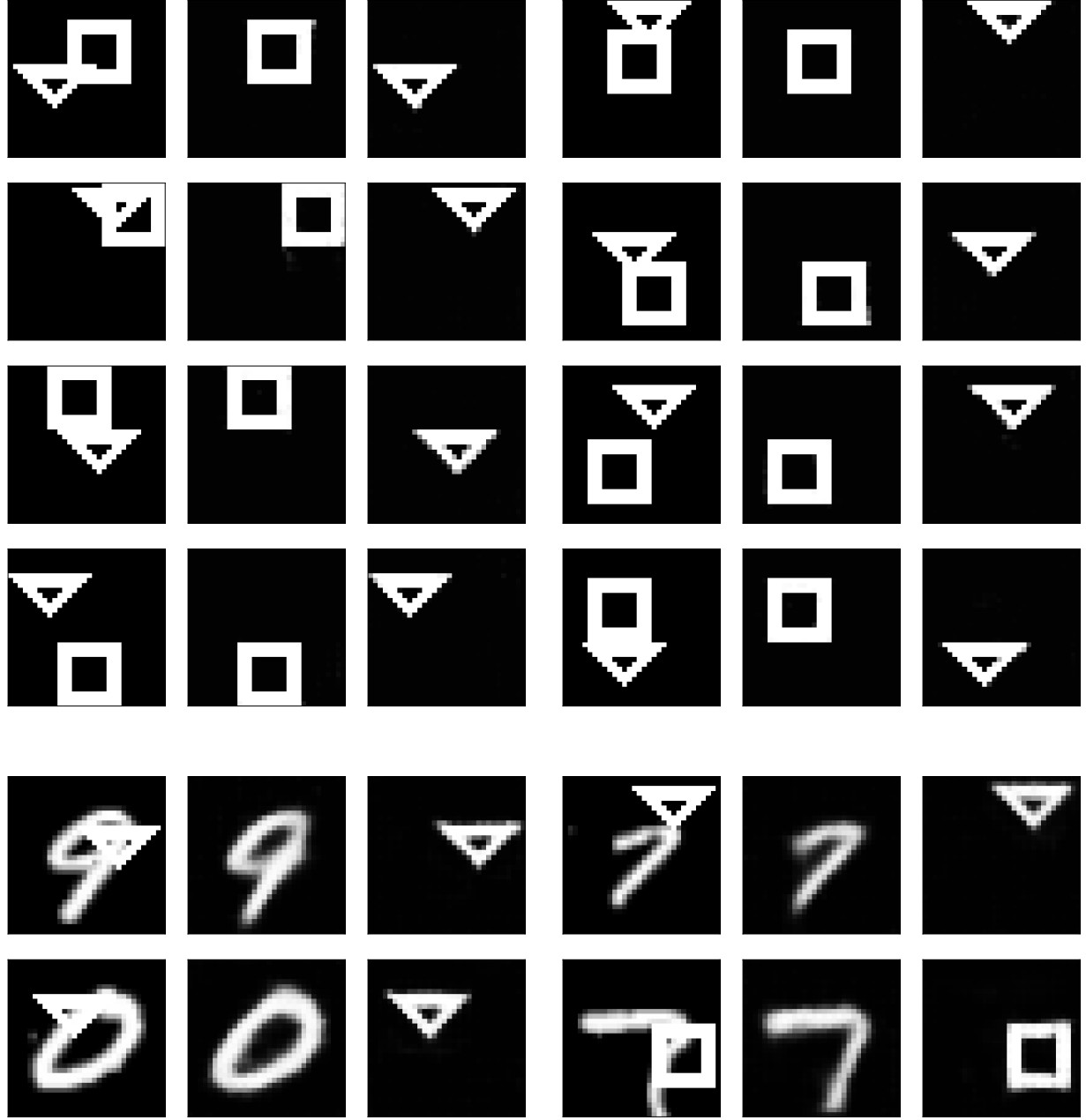

Figure 14: Investigating object-centricity of the latent features in the CAE. **Top**: 2Shapes, **bottom**: MNIST&Shape dataset. **Columns 1 & 4**: input images. **Columns 2-3 & 5-6**: object-wise reconstructions. By clustering features created by the encoder according to their phase values, we can extract representations of the individual objects and reconstruct them separately.

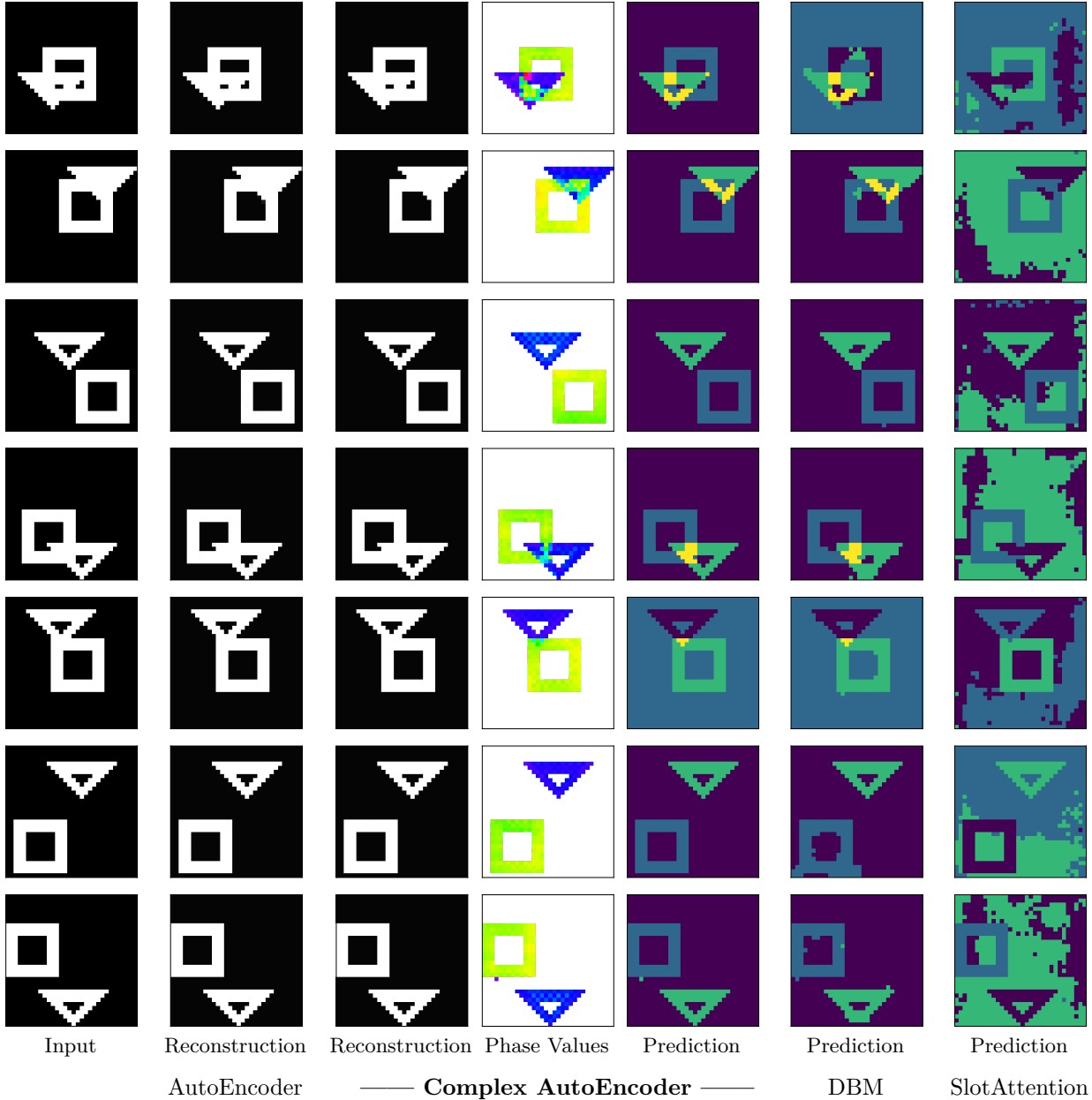

Figure 15: Visual comparison of the performance of the Complex AutoEncoder, its real-valued counterpart (AutoEncoder), the DBM model and the SlotAttention model on random test-samples from the 2Shapes dataset.

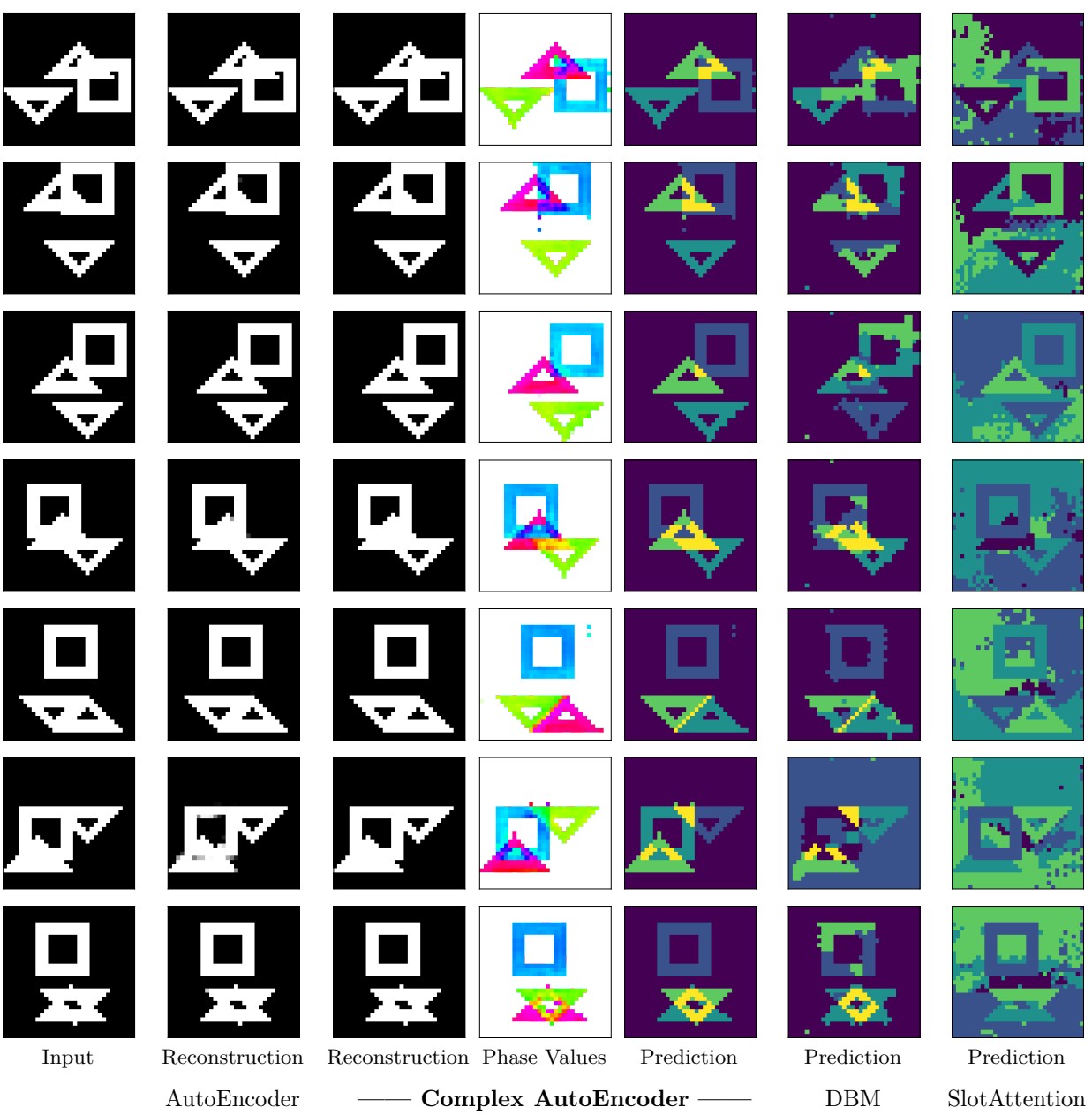

Figure 16: Visual comparison of the performance of the Complex AutoEncoder, its real-valued counterpart (AutoEncoder), the DBM model and the SlotAttention model on random test-samples from the 3Shapes dataset.

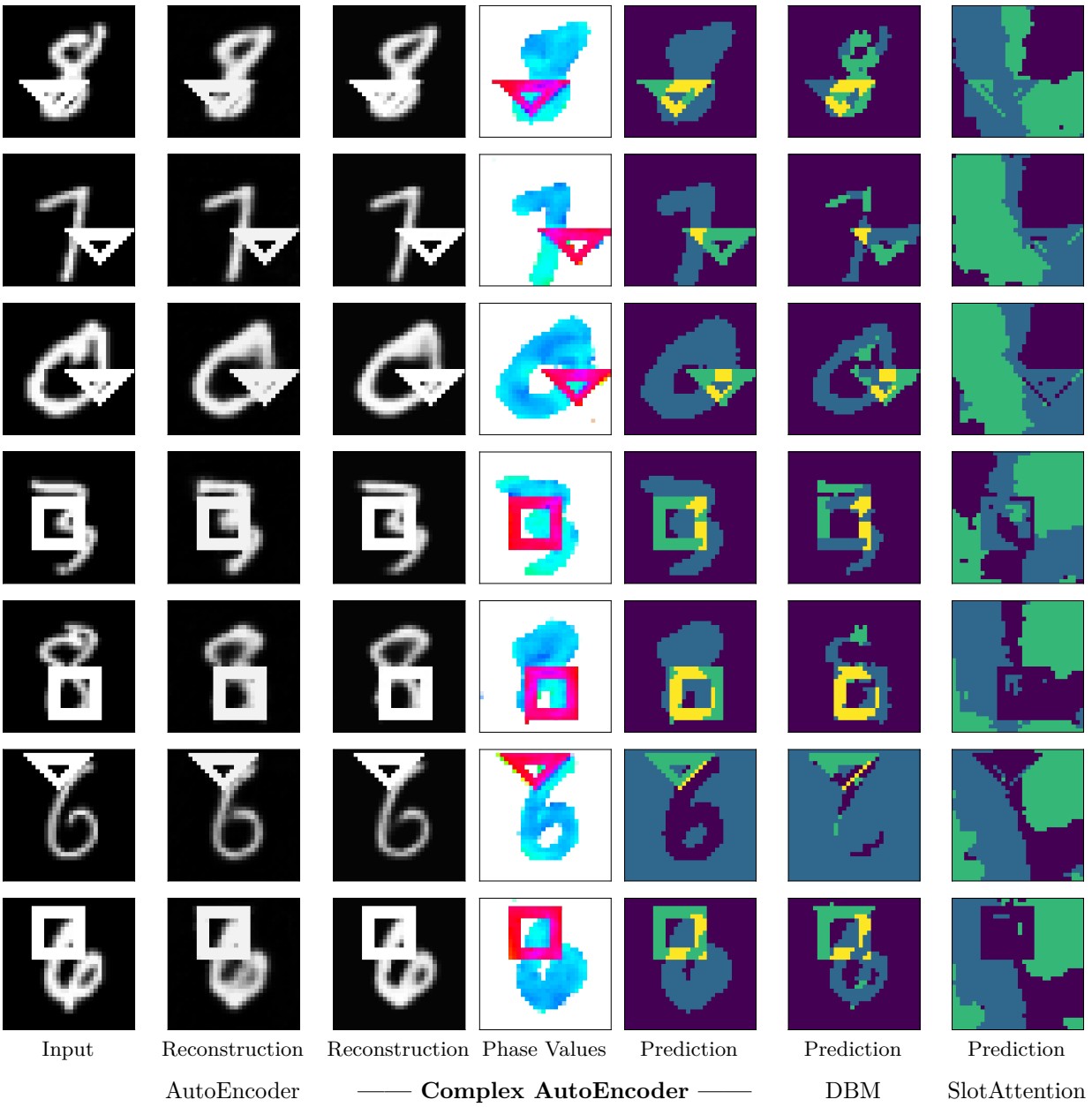

Figure 17: Visual comparison of the performance of the Complex AutoEncoder, its real-valued counterpart (AutoEncoder), the DBM model and the SlotAttention model on random test-samples from the MNIST&Shape dataset.

