# OpenReview forum: "Complex-Valued Autoencoders for Object Discovery"
_TMLR — Accepted by TMLR_

### Review · Reviewer_tMFe · 2022-10-02

**Summary Of Contributions:**

This work proposes using complex-valued auto-encoders to automatically discover objects in the images through finding the aligned phases in the output vectors. The authors claim that this method achieves comparable performance to the earlier method SlotAttention, but uses much less computation time during training. After showing results on multiple manually constructed datasets, which contain several shapes or one shape plus one digit, the paper shows that this method reaches cleaner object separation and low reconstruction loss.

**Broader Impact Concerns:**

The authors have already addressed this in the paper and there is no clear negative societal impacts from this method.

**Requested Changes:**

See the points in the weaknesses. Addressing the first point would be required. Other points can strengthen the work.

**Strengths And Weaknesses:**

Strengths:
-	The fact that the correlations between the phases can directly discover different objects is quite inspiring and interesting.
-	The results shown on the datasets tested by the authors also support their claim that the reconstruction of the input images are successful and the discovery of the objects also works well.
-	The paper is well written and clear in its logics.

Weaknesses:
-	The biggest issue, and in fact a critical one, is that the datasets evaluated in this dataset are too simple and easy. This is especially concerning as the SlotAttention method, which is (or was) the state-of-the-art method in this object discovery problem, was proposed and tested on datasets that are far more complicated, realistic, interesting, and useful, like CLEVER6 and Multi-dSprites. Not mentioning that SlotAttention was published already 2 years ago and there should be even better methods (like MONET) than it. Although the authors mentioned this as the limitation of this work in the end of this paper, it still makes the value of this work significantly less. So in my opinion, it would be critical to show that this method can at least get reasonable results on the datasets SlotAttention was tested on in its paper.
-	The use of datasets that are simpler than what were used in SlotAttention’s paper creates another issue for a fair comparison to that method, as it was never tested on these datasets and hyperparameters may not be fine-tuned for them.
-	The right part of Figure 2 makes me wonder whether there will be an upper limit in the number of objects that can be discovered in the image. In fact, from this visualization, I would think that having more objects in the images will make the performance on the object discovery worse, is this true? How many objects can this method handle without the loss of performance?
-	This is a smaller issue, can this method discover and separately identify multiple instances of the same object shape? The current used datasets do not seem to have this situation.

---

> ### Author Response · Authors · 2022-10-09
> **Reply**
>
> Thank you for the constructive review. We are very happy to see that you find our paper “well written and clear in its logics” and will reply to your comments below:
>
> ### Simple Datasets
> We carefully selected TMLR as the journal for our submission due to its focus on original papers with sound empirical validation but potentially modest significance. Although we fully agree that this work could be improved further if it showed state-of-the-art results on more complex datasets, we believe that the paper in its current form rigorously supports all claims with accurate evidence and that it is interesting for people in TMLR’s audience and others in the object-centric representation learning community - and thus fits well with the two evaluation criteria of TMLR. As such, we highly appreciate that the reviewer regards our work as “quite inspiring and interesting”; and the claims within our paper to be supported by the results.
>
> We see our paper as an important initial step to explore alternative object-centric representation methods. While this alternative is not immediately competitive with the state-of-the-art on more realistic datasets, we still believe that it is worth exploring and worth sharing with the community. As described in the paper, there are two main limiting factors that prevent the CAE from being applied to more realistic data: (1) the number of objects that the model can represent at once and (2) the scaling to RGB data. More research is required to overcome these limitations; and with this paper, we hope to get this research started.
>
>
> ### Fair comparison to SlotAttention
> We spend considerable effort to fine-tune SlotAttention’s hyperparameters for the settings in which we test it, as outlined in Appendix C.3. In fact, without developing the presented decoder architecture, the SlotAttention model would not have achieved competitive performance to the CAE in terms of ARI-BG score on the 2Shapes and 3Shapes datasets.
>
>
> ### Maximum number of objects that can be represented
> For now, the CAE is quite restricted in the number of objects that it can represent at once. As shown in the “Dataset Sensitivity Analysis” paragraph of the experiments, the model fails on the 4Shapes dataset, which contains four objects in each image. Thus, it seems that the model can only reliably represent three objects at once. We believe that overcoming this limitation is one of the most pressing directions for future research, as it currently restricts the CAE’s applicability to more realistic datasets.
>
>
> ### Same object shape
>
> We thank the reviewer for this suggestion and are happy to report that the CAE can indeed handle multiple instances of the same object shape. To test this, we ran an experiment on a new dataset in which each image contains two triangles of the same orientation. The CAE achieves the following performance on this dataset (mean $\pm$ sem across 8 seeds):
>
>
> MSE: 4.441e-06 $\pm$ 4.046e-06
>
> ARI+BG: 0.879  $\pm$ 0.019
>
> ARI-BG: 0.948 $\pm$ 0.045
>
> Given the high ARI-BG score that the model achieves, we conclude that it can handle datasets in which the same object shape appears twice within an image.
>
> We will add this experiment to the revised version of the paper.

---

> > ### Comment · Reviewer_tMFe · 2022-10-14
> > **Response**
> >
> > I thank the authors for their response. I agree that the requirement for results on the more advanced datasets is unnecessary for acceptance to this journal. I apologize for making that requirement in my initial review, which was made before I learned that TMLR focuses on technical soundness.
> > I am glad that the authors provide additional experiments showing that they can distinguish different instances of the same shape. The model still has some constraints, but the results are well-supported, and the algorithm is inspiring. I would especially recommend the authors work harder to allow representations of more objects.
> > I would now recommend accepting the paper.

---

### Review · Reviewer_V5RU · 2022-10-04

**Summary Of Contributions:**

This paper presents a novel avenue to learn object-centric representations, by leveraging complex valued neural networks. More precisely, they set up an Autoencoder with complex activations throughout, and use the phase at the output of the decoder as an indicator for feature binding (i.e. an implicit segmentation per object as a function of phase).

They present results on toy grayscale datasets (a clear limitation of the approach, as I will cover below), comparing against DBM (the only previously similar instantiation of such an idea) and SlotAttention (current baseline in slotted object representation learning, near SOTA enough).

Overall, the results are interesting and convincing of the approach’s potential, but in its current state this is a very preliminary line of research. This is both an issue and an opportunity, as there are clearly defined improvements that need to be made to this line of research, which this paper can help publicize and create a community for.

Given the “TMLR two questions” for submissions, I would answer yes to both and lean towards acceptance, but some of the work I would like to see might be too large for a minor revision to accept the paper at this time.

**Requested Changes:**

1. Comment on the grayscale limitation and on the difficulty of scaling to RGB earlier in the paper. I would also expect a deeper discussion of why an “hybrid” solution like I presented above would not be satisfactory.
2. Comment / demonstrate the model’s ability to learn useful object-centric “representations”, i.e. have something that uses the latent codes for an object-driven task. This might be necessary to bring this paper to an effective form, which might be too much work for a minor revision.
3. Comment on the failure of SlotAttention, or modify the presentation to more clearly indicate what it does well or not.
   1. I would have expected it to work, so it is important to explain why this failed. Right now I am not reassured that the model was applied “well-enough”.
   2. For example, you report ARI+BG first, even though the field uses ARI no BG as a metric (this is usually because backgrounds do not have a single “correct” segmentation, so failing to find the one that exists in the segmentation labels isn’t that fair)
4. I would recommend moving Figure 6 to Appendix to make space for some extra datasets to go into the main text (e.g. take Figure 12 in the Appendix, we need more phase diagrams!)

**Strengths And Weaknesses:**

1. The paper does a good job of presenting the approach and motivation, and it explains its choices very well.
   1. However I would say that by Section 2 I had heard the motivation enough times and I would recommend trimming it down. It is quite clear where the idea comes from, so the paper would benefit from getting to the point faster. There is quite a lot of redundancy from the abstract to the start of Section 2.
   2. Sections 3.1 - 3.3 cover the model choices well, although once again I probably would have compressed them a bit as all of them appear really sensible and “simple” (not a bad thing!)
   3. Section 3.4 is nice to have (albeit slightly trivial), but you do not emphasize why/when having such an equivariance would help / hurt?
      1. However, Figure 6 and the associated section in results isn’t particularly useful, and I would cut it entirely or move it to the Appendix. Again, I am not sure this is a particularly “useful” example to demonstrate.
2. The fact that the model is constrained to grayscale isn’t made clearly and early enough.
   1. This is the main limitation of the method, and having this discussed only briefly in the last section and Appendix C.1 is not sufficient. It only comes in Section 4.1, as up till then I assumed you would cover extra color datasets.
   2. It also feels to me like your answer to why RGB is difficult isn’t that convincing.
      1. Shouldn’t the model be able to output RGB yet be forced to use phase as implicit segmentation still?
      2. Your example is very particular, instead how would you want to represent a yellow object?
      3. Why can’t you use a single phase channel and 3 real activation channels? I understand that this breaks the mental model of “complex numbers everywhere”, but your model already handles the real and complex part nearly independently, so that connection is strenuous at best? I would have expected a more innovative discussion of how to make this work.
   3. Given this constrained your choice of datasets, you should discuss it earlier, and perhaps indicate that the datasets considered might actually be more challenging to extract objects out of given their lack of color separation.
3. The model’s behavior on the dataset assessed is interesting and convincing enough of the usefulness of the method and would allow people to build upon them.
   1. Figure 4 and 5 are the ones I found most informative, especially with the phase diagram and the object-centric reconstructions.
      1. Could we have more of these?
      2. However, where is the background encoded in the phase diagram?
   2. In Figure 3, I did not understand the yellow area or why you’d remove them before evaluation? What is the “correct” output?
4. However, the model’s ability to be a “good” object-centric method is not thoroughly investigated in this current draft.
   1. For example, one would need to show that training on a varying number of objects allows the model to handle both more and fewer objects. You try to do that with <3Shapes, but this isn’t hard enough, e.g. what if you train on 1, 3, 5, can you generalise to 2 and 6?
   2. Can you leverage the “phase separation” at the representation level? If this is a representation learning method, what matters is how useful the latent code is, not its reconstruction ability / quirknesses.
      1. Can you decode the features of individual objects well?
      2. Are features specializing into subsets with phase as well? Or is that only “clean” at the output level?
   3. How does this change when varying the number of objects in the input? Is it stable or does it decay catastrophically past a certain “input load”?
      1. Looking at the phase plots in Figure 4, it feels like there would be a natural “maximum number of objects” that one can attend to.
      2. As an aside, this isn’t intrinsically bad, as this is sometimes used in human visual short term memory to explain the limited memory capacity (e.g. “Magical number 4 in short-term memory” by Cowan, although don’t take this number literally). In the brain the phasic load is probably within one gamma cycle, but the principle stays the same.
      3. If there was a way to have control over which ones are attended to, or randomizing them, this could be worked around.
   4. Having to do a post-hoc clustering isn’t particularly elegant, especially since areas that overlap appear like another object (my reading of the phases in Figure 4).
      1. How many “objects clusters” did you choose in Figure 4 right-most? The phase diagram might indicate you would need 4, how would you select this automatically?
      2. Could you define a task which would leverage the fact that the “segmentation” information is present in a continuous fashion? Having a continuous indicator is actually quite nice if that allows later stages of processing (maybe using “normal” neural networks) to do a better job than if that wasn’t encoded/available at all?
   5. I also did not really understand why you had to do the scaling operation for small magnitudes.
      1. Wouldn’t this have allowed you to detect “unused” dimensions and prune them away?
      2. Where would they have fallen in the phase diagram of Figure 4 before / after this modification?
5. The performance of SlotAttention on your datasets is surprisingly bad, and I would not have expected such a gap.
   1. This would need to be discussed more deeply.
      1. Is it due to the grayscale nature of the data?
      2. Is it that the objects are slightly too large? (but in that case you could have made them smaller, if SlotAttention has an intrinsic prior over object size due to the number of effective slots after its convolutional stack)
   2. In effect, the SlotAttention predictions in Figure 3 just make me feel like something went wrong, because there is no reason to expect this kind of failure mode (given such harsh binary inputs, another failure would have been to mix the 2 shapes into one slot).
   3. What is its behavior on the 2Shapes-randBG dataset?
6. I understand that the DBM baseline was the closest to this current work, but I am not sure of how much value it brings. Nobody uses them anymore for many reasons, which you also point out, so improving on that is not a surprising fact?
   1. It is really apparent in the Appendix that a lot of work has been put in reproducing them, so this is quite unfortunate, but I am not sure this adds much value to the paper.
7. Nits / typos
   1. In section 3.1, when mentioning f_out, add a mention that you found this helpful and already ablate it in Table 2. It feels slightly overkill when one gets to this point of the manuscript and I was about to ask for an ablation.
   2. In section 3.2, last paragraph, you use z_2 and w_2 but these should be bold symbols as they are vectors.

---

> ### Author Response · Authors · 2022-10-09
> **Reply (1)**
>
> Thank you for the thorough and constructive review. We are very happy to see that you find our results “interesting and convincing” and will answer your questions below. We grouped our answers by related topics and indicate the numbering under which the corresponding question can be found in the review.
>
> ### Masking of small magnitudes
> *(4.5) Why do we scale small magnitudes during evaluation?*
>
> We rescale features with small magnitudes to avoid unnecessary noise during the evaluation. The closer a feature is to the origin, the less its phase will influence how it will be processed by the network, and as a result, the network has no incentive to assign anything but random phase values to these features.
>
> To evaluate how well the phases separate the objects in the scene, we need to evaluate them as independently of the magnitudes as possible. To achieve this, we normalize most features such that they will fall on the unit circle. If we did not treat features with small magnitudes separately, however, these would be projected out onto the unit circle, essentially amplifying their noisiness. Instead, we rescale them such that they remain relatively close to the origin - ensuring little noise in the evaluation, while relying minimally on the magnitudes.
>
> As a result of this rescaling procedure, features with small magnitudes are usually assigned their own cluster by the k-means algorithm. Thus, this method will behave similarly to a pruning procedure, while being slightly more flexible as it does not enforce a hard threshold.
>
>
>
> *(4.5.2) Where would the small magnitudes have fallen in the phase diagram of Figure 4?*
>
> These features are actually present in the Figure - they all lie on the origin. By applying the scaling operation, we essentially ensure that they remain there, close to the origin, instead of spreading them out onto the unit circle together with all other features - which would only add noise to the clearly separated phases of the objects.
>
>
>
> *(3.1.2) Where is the background encoded in the phase diagram?*
>
> In our main experiments, the background has a pixel value of zero, which generally results in small magnitudes in the features that reconstruct it. Thus, the corresponding phase values are rescaled during evaluation. To highlight this in the phase diagrams, we set the alpha values for each pixel in the plot equal to the rescaled magnitude - resulting in a white area for features with small magnitudes.
>
> Note, that this does not mean that the network requires a black background to achieve a good separation between the foreground and background, as highlighted by the 2Shapes-randBG dataset.
>
>
> ### Grayscale vs. RGB
> *(2) & Requested Change (1)*
>
> In its current form, the CAE cannot be applied to RGB data. This restriction is largely due to its evaluation method rather than the model itself. In fact, it is possible to simply scale the input/output dimensions to three channels and train the CAE on RGB data - the problem is that it is unclear how to evaluate the resulting phase values in a convincing way. This problem arises from the rescaling of features with small magnitudes. If one of the RGB channels is assigned a small magnitude, this rescaling leads to a trivial separation of the objects, as described in Appendix C.1.
>
> To overcome this limitation, we could try to adjust the final layer in the network to output a single phase, while outputting three magnitudes independently, as suggested by the reviewer. One possible challenge for this approach might be the lack of a training signal for the phase channel. Since we train the network in an unsupervised way, we only apply a loss on the magnitudes and solely rely on the close connection between the magnitudes and phases to enable the latter to learn a good object separation. It remains an open question how one could implement a similarly close connection when there is no one-to-one relation between the magnitude and phase channels.
>
> We agree with the reviewer that this is one of the major limitations of the CAE model as proposed in this paper. Therefore, we will follow their suggestion to highlight this limitation earlier in the paper and to include a more thorough discussion of it (as outlined above).

---

> > ### Author Response · Authors · 2022-10-09
> > **Reply (2)**
> >
> > ### Object-centric representations
> > *(4.2) + Requested Change (2)*
> >
> > *Can you leverage the “phase separation” at the representation level?*
> >
> > Yes, we can leverage the phase separation at the representation level, and we can use this to decode the features of individual objects as shown in Figure 5 of the paper. For this figure, we input an image with two objects into our model and encode it using the encoder $f_{\textrm{enc}}$. Then, we separate the latent representations based on their phases by using the clustering procedure as described in section 3.5. As a result of this unsupervised procedure, we get two feature vectors, one for each object, where each vector is filled with the features belonging to one of the phase clusters. As shown in the figure, we can use those feature vectors to accurately reconstruct the individual objects. Thus, we conclude that the latent representation of our model is object-centric: the representations of the individual objects can be clearly separated from one another, and they contain all the necessary information for the individual objects to be reconstructed accurately.
> >
> > We agree with the reviewer that including more images to highlight this capability of the CAE would strengthen the paper, and will therefore add more reconstruction images to the Appendix.
> >
> >
> > ### SlotAttention performance
> > *(5) & Requested Change (3)*
> >
> > *Why does SlotAttention fail on the MNIST&Shape dataset?*
> >
> > We have put significant effort into ensuring that our implementation of SlotAttention is correct and that the result we present is a true failure mode of the model. For one, our implementation closely follows the original implementation, and we have validated that it achieves the same performance on both the Multi-dSprites and Tetrominoes datasets as reported in the original paper. Our implementation’s good performance on the 2Shapes and 3Shapes datasets further assures us of its correctness. Additionally, we tried to ensure a fair comparison by testing a range of settings for the SlotAttention model as outlined in Appendix C.3.
> >
> > Following the reviewer’s suggestion to investigate the different factors that lead to SlotAttention’s failure on the MNIST&Shape dataset, we ran four additional experiments. First, we trained SlotAttention on a version of the MNIST&Shape dataset in which the MNIST digits are downsized to 16x16 pixels. On this dataset, SlotAttention fails to separate the two objects. Second, we trained SlotAttention on a version of the MNIST&Shape dataset in which the MNIST digits are of the original size, but binarized. Again, SlotAttention fails on this dataset. If we combine the two modifications and create a dataset with smaller, binarized MNIST digits, SlotAttention finally starts to separate the objects. However, it still does not perform perfectly - in some cases, it splits the digits into two objects, e.g. by splitting an eight into two separate circles. Last but not least, we applied SlotAttention on the 2Shapes-randBG dataset as suggested by the reviewer. Here, SlotAttention performs perfectly for half of the seeds; and largely fails for the other half. Interestingly, the seeds that fail in terms of object discovery performance achieve a better reconstruction performance.
> >
> > The table below lists all results for these experiments, highlighting the mean $\pm$ sem performance across 8 seeds.
> >
> > | Experiment                    |                  MSE Loss |             ARI+BG |            ARI-BG |
> > |:----------------------------- | -------------------------:| ------------------:| -----------------:|
> > | small MNIST&Shape             |  0.004495 $\pm$ 0.0004804 |  0.203 $\pm$ 0.086 | 0.365 $\pm$ 0.088 |
> > | MNIST&Shape binarized         |  0.030710 $\pm$ 0.0011260 |  0.081 $\pm$ 0.018 | 0.153 $\pm$ 0.038 |
> > | small MNIST&Shape binarized   |  0.010250 $\pm$ 0.0007853 |  0.380 $\pm$ 0.108 | 0.851 $\pm$ 0.022 |
> > | 2Shapes randBG                | 1.190e-05 $\pm$ 2.235e-06 |  0.538 $\pm$ 0.174 | 0.604 $\pm$ 0.166 |
> > | 4 working seeds: 2Shapes randBG | 1.526e-05 $\pm$ 3.598e-06 | 0.997 $\pm$  0.001 | 1.000 $\pm$ 0.000 |
> > | 4 failing seeds: 2Shapes randBG | 8.539e-06 $\pm$ 1.685e-06 |  0.078 $\pm$ 0.032 | 0.208 $\pm$ 0.154 |

---

> > > ### Author Response · Authors · 2022-10-09
> > > **Reply (3)**
> > >
> > > Based on these observations, we believe that SlotAttention fails on the MNIST&Shape dataset due to two factors: (1) the MNIST digits are too large to be covered by SlotAttention’s receptive field. Only when the MNIST digits are downsized, can the SlotAttention model learn to separate the objects; (2) the SlotAttention model performs poorly on grayscale values, as highlighted by its results on the non-binarized MNIST&Shape datasets and the 2Shapes-randBG dataset. We hypothesize that this is due to the alpha mask that is used in the spatial-broadcast decoder: the SlotAttention model might learn to use this mask to reconstruct the precise grayscale value rather than to correctly merge the reconstructions from the individual slots. This problem would be naturally circumvented by RGB data.
> > >
> > > We agree with the reviewer that it is important to provide an explanation for the failure of the SlotAttention model on the MNIST&Shape dataset. Therefore, we will include the full evaluation of the four experiments described above, as well as the resulting discussion, in the Appendix of the revised paper.
> > >
> > >
> > > ### ARI+BG vs ARI-BG
> > > *(Requested Change 3.2)*
> > >
> > > We agree with the reviewer that ARI-BG is often the more useful metric, and thus report ARI-BG scores for all experiments. The ARI+BG score is only meaningful in combination with a high ARI-BG score, as mentioned in the “Metrics” paragraph of Section 4.1. Because of this, we only report it in some experiments to highlight that the CAE can accurately separate the foreground from the background, while SlotAttention cannot.
> > >
> > >
> > > ### Global Rotation Equivariance
> > > *(1.3) & Requested Change (4)*
> > >
> > > *Why would having such an equivariance help?*
> > >
> > > We believe that the global rotation equivariance of the CAE is relevant for two reasons. Firstly, as described in Section 3.4, it improves the model’s biological plausibility. In the brain, the point in time at which a stimulus is perceived should influence when the stimulus is processed, but it should not influence how it is processed. Equivalently, in the CAE, the input phase should influence the output phase, but not the output magnitudes. Secondly, only if the model is equivariant w.r.t. global rotations can it truly bind features “by synchrony”, i.e. based on their relative phase differences to one another. Without this equivariance property, the model could use the absolute phase values to learn to separate objects and could thus learn specialized representations for each object. However, we want the model to learn a common representational format for all objects instead. The SlotAttention model is permutation equivariant w.r.t. the order of the slots to achieve this; in our case, we need the model to be equivariant w.r.t. global rotations to achieve a similar behavior.
> > >
> > > We will include this discussion in the revised version of the paper. Additionally, since we believe that this property is quite crucial for the CAE, we would like to keep Figure 6 in the paper.
> > >
> > > ### Clustering for Evaluation
> > > *(4.4) How many object clusters did you choose? How would you select this automatically? Could you evaluate the continuous object assignment directly?*
> > >
> > > In our experiments, the number of object clusters was always chosen to correspond to the true number of objects in the image. Since this is rather restrictive, we hope to develop better evaluation methods in the future that do not require $k$ (i.e. the number of clusters) to be set and known in advance. When trying to achieve this by using alternative clustering procedures, it might indeed be problematic that the model assigns intermediate phase values to the overlapping areas, as pointed out by the reviewer. However, one could also interpret this behavior as a desirable property of the model, as it accurately expresses the model’s uncertainty in these areas. Thus, we agree with the reviewer that it would be more elegant to evaluate the continuous object assignments directly, as this could remove the requirement to set $k$ in advance while making use of the uncertainty expressed by the model. We hope to address this in future work; for now, we decided to create discrete object assignments using post-hoc clustering to allow for a straightforward comparison to previous approaches using established evaluation metrics.

---

> > > > ### Author Response · Authors · 2022-10-09
> > > > **Reply (4)**
> > > >
> > > > *(3.2) Why do you remove the yellow area before evaluation (Fig. 3)?*
> > > >
> > > > As noted by the reviewer, the phase values for areas in which the objects overlap fall in between the phase values of the individual objects. Since these intermediate phase values might lead to a lower performance of the k-means algorithm, we remove areas in which objects overlap before applying our clustering procedure - resulting in the yellow areas in Figure 3.
> > > >
> > > > We believe that this is the best way to ensure a fair comparison between methods. In the grayscale case, there is no “correct” output for these overlapping areas, since it remains unclear which object is in the foreground and which one is in the background. Thus, we need to remove these areas at some point during evaluation, either before or after the clustering procedure. Ideally, we would want to remove them after the clustering procedure to minimize the amount of prerequisite knowledge that we need to assume for the evaluation procedure. However, if we were to do that, we would essentially reward the model if it were to assign these undetermined areas to one of the objects, as this would lead to the cleanest clustering results. We believe that this is not desirable: we should not penalize a model for accurately expressing the (irreducible) uncertainty that it faces. Thus, we decided to remove the areas in which objects overlap before applying the clustering procedure.
> > > >
> > > >
> > > > ### Limitations
> > > > *(4.3) What is the maximum number of objects that the CAE can handle?*
> > > >
> > > > As shown in the “Dataset Sensitivity Analysis” paragraph of the experiments, the model fails on a dataset that contains four objects in each image. Thus, it seems that the model can only reliably represent three objects at once. We believe that overcoming this limitation is one of the most pressing directions for future research, as it currently restricts the CAE’s applicability to more realistic datasets.
> > > >
> > > >
> > > > *(4.1) Can the CAE generalize to different numbers of objects?*
> > > >
> > > > With the $leq$3Shapes dataset, we train the model on images with three objects, and show that it generalizes to fewer objects. Given the above limitation on the number of objects that the model can represent, this is currently the most challenging setting that we can create to test the generalization capabilities of the model to a smaller number of objects.
> > > >
> > > > Based on the reviewer's comments, we also tested whether the CAE generalizes to more objects. For this, we train the model on the 2Shapes dataset (with one square and one triangle per image) and test it on a dataset in which each image contains 3 shapes (one square and two triangles of the same orientation). We find that the performance drops substantially in this setting, with the model achieving the following results (mean $\pm$ sem across 8 seeds):
> > > >
> > > > MSE:         3.979e-02 $\pm$ 9.438e-04
> > > >
> > > > ARI+BG:   0.844 $\pm$ 0.008
> > > >
> > > > ARI-BG:    0.758 $\pm$ 0.007
> > > >
> > > > Thus, it seems that the CAE does not generalize to more objects than observed during training. We believe that this is due to the way that the CAE learns to assign different phases to objects: it generally pushes these phases as far apart from one another as possible. When trained on two objects, the model thus assigns directly opposite phases to the two objects - leaving no space for the third object to be represented.
> > > >
> > > > We will include the above experiment and a discussion of the ensuing limitation in the paper.
> > > >
> > > >
> > > > ### Comparison to the DBM model by Reichert et al. (2014)
> > > > *(6) Why do we include the comparison in the paper?*
> > > >
> > > > As pointed out by the reviewer, the DBM model by Reichert et al. (2014) is the closest existing work to the proposed CAE. We therefore believe that it is important to include a comparison to this model in the paper, as this is the only way to highlight the improvements that our model brings about.
> > > >
> > > >
> > > > ### Paper presentation / Nits / typos
> > > > *(1) & (7)*
> > > >
> > > > We thank the reviewer for these suggestions and will do our best to incorporate them into the revised paper.
> > > >
> > > >
> > > > *(3.1.1.)*
> > > >
> > > > We include phase diagrams similar to Figure 4 in Figure 12 of the Appendix. We will also include more reconstruction results similar to Figure 5 as suggested by the reviewer.

---

> > > > > ### Comment · Reviewer_V5RU · 2022-10-12
> > > > > **-**
> > > > >
> > > > > Thanks a lot for your thorough response and for all the great new experiments you ran in a very short amount of time.
> > > > >
> > > > > I am happy with how my proposed changes where handled or implemented, and would now be happy to support the acceptance of the paper in this form.
> > > > >
> > > > > Comments:
> > > > > - Thanks for the explanation of the masking of the small amplitudes. As you mentioned, this is a rather specific problem to tackle and I feel like your choices were very warranted. Sorry I did not see the dots on the origin, this makes more sense now! I also like the idea of letting the model use the uncertainty of object-assignment through phases being allocated differently, that's a great observation.
> > > > > - On the RGB question, I believe this is totally fine to leave this to further work given these extra complexities. With the earlier mention this will then fully validate the TMLR expectations as well on my end!
> > > > > - Indeed you are right that the individual reconstructions already provide a good signal for the useful of the latents, sorry I overlooked that aspect! You may already have done that, but I missed the "phase clustering" being done on the latent space (it is obvious in hindsight) so perhaps worth strengthening this point in the main text too. For further work, I feel like "just" adding a feature decoder post-hoc would be a nice extra table (especially if you're being ultra clear about the encoder being frozen / stop-gradient'd), but this isn't necessary to prove your point. The fact that the model fails to handle extra objects is a bit disappointing, but as you mention there is an expectation and understanding of why, so probably something that can be flagged explicitly for further work?
> > > > > - Thank you for the extra assessment of SlotAttention. What you find makes sense to me now, and as you mention would be very useful for others practitioners to be aware of. The seed instability on 2Shapes-randBG is particularly interesting too!
> > > > > -  For ARI-BG, I probably would not have raised that if you reordered the columns in Table 1 actually :) So it might be as simple as moving ARI+BG to the last column.
> > > > > - No disagreement on my part on the choice to keep the DBM + Global rotation equivariance sections in given your preferences.

---

### Review · Reviewer_NrE9 · 2022-10-08

**Summary Of Contributions:**

The paper proposes the Complex AutoEncoder (CAE) for unsupervised learning of distributed object-centric representations. In contrast to state-of-the-art methods that explicitly represent each object as a separate slot in the model architecture, the proposed method explores using synchronization among distributed neural activations for object discovery. This is motivated by the temporal correlation hypothesis in neuroscience, that posits that the firing rate of a biological neuron encodes the presence of some feature, and the synchronous firing patterns among multiple neurons indicate the binding of features. To that end, the proposed CAE uses complex-valued activations, where the phase of the complex value can indicate synchronization. The paper makes some design choices to encourage the emergence of synchronization. The resulting model has no iterative procedure and is simpler compared to slot-based methods. After training, a clustering procedure is applied to the phase values to read out the synchronized groups, which are shown to correspond to objects. Experiments on a few simple datasets show that the proposed method outperforms a previous work that uses complex values at test time by a large margin in terms of reconstruction quality and object discovery. When compared to Slot Attention, a state-of-the-art slot-based method, the proposed CAE succeeds in decomposing images containing a small shape and a large MNIST digit where Slot Attention completely fails. CAE also does not over-segment the background, and is much faster to train than both baselines. The paper also theoretically proves and visually demonstrates that CAE is equivariant to phase shifts in the input.

**Broader Impact Concerns:**

sufficiently addressed

**Requested Changes:**

Below are some minor comments that I hope the authors could address.
- It looks like the dot ($\cdot$) stands for element-wise multiplication. Can you state that explicitly? Also in Eq 8-9, does $w^\top \cdot z$ mean inner product, element-wise product, or matrix multiplication?
- I didn't find the explanation or ablation for adding a bias term on the magnitude ($b_m$ in Eq 3). I understand that this may be just a natural choice to make the network more flexible. Can you clarify?
- Is the $f_w$ in Eq 2 and Eq 4 shared (similar with $b_m$)? If so, can you elaborate on why they are shared?
- The term "maximally distant phase" is used a few times. It would be better to include a precise definition for that.
- I didn't fully understand how to "fine-tune the decoder to reconstruct separate objects from the individual clusters". How do you know which object to reconstruct?
- It is mentioned that the background cluster tends to have low magnitude. Is this general or just because the background is black? What happens when you use random sampled background (2Shapes-randBG dataset)?

**Strengths And Weaknesses:**

- Strengths
    - The paper proposes the first end-to-end trainable complex-valued neural network for unsupervised object discovery. The idea is quite interesting as it is inspired by how biological neural networks solve the binding problem.
    - The qualitative results convincingly show that the model can indeed output similar phase values for the same object.
    - The MNIST&Shape dataset seems particularly challenging for slot-based methods, as it contains objects of different sizes. It is quite impressive that the propose model achieves almost perfect foreground segmentation while Slot Attention completely fails.
    - There is ablation study that validates the design choices.
    - The paper is generally well written. The idea of using complex values and the network design are well motivated.
    - The paper makes modest claims, and clearly states its limitations.
    - Related work seems sufficiently discussed.
    - Implementation and dataset details are provided in Appendix.
    - The experiments are run with multiple seeds.
- Weaknesses
    - I didn't find any major weakness. However, some clarifications (listed in Requested Changes) could be helpful to better understand the paper.

---

> ### Author Response · Authors · 2022-10-09
> **Reply**
>
> Thank you for the constructive review. We are very happy to see such a positive evaluation of our work and will reply to your question below:
>
>
> **It looks like the dot ($\cdot$) stands for element-wise multiplication. Can you state that explicitly? Also in Eq 8-9, does $w^t \cdot z$ mean inner product, element-wise product, or matrix multiplication?**
>
> We thank the reviewer for pointing out these inconsistencies in our notation. We agree that this should be improved, and we will clarify it in the revised version of the paper as follows:
>
> So far, we use the dot ($\cdot$) interchangeably for both element-wise and scalar multiplications. To improve upon this, we will replace the dot with a circle ($\circ$) for element-wise multiplications, and clearly state the meaning of both symbols.
>
> In Eq. 8-9, the product between $w$ and $z$ is supposed to be an inner product. We will make this more explicit as well by replacing the current formulation with the $\langle, \rangle$ notation and by stating that this notation stands for the inner product.
>
>
> **I didn't find the explanation or ablation for adding a bias term on the magnitude ( in Eq 3). I understand that this may be just a natural choice to make the network more flexible. Can you clarify?**
>
> We introduced the magnitude bias to create a more consistent overall formulation, which applies a bias on both the magnitude and the phase. However, this term is not strictly needed, as it is effectively canceled out by the BatchNormalization that we apply in Eq. 5. As a result, we experience that the magnitude bias does not have a noticeable impact in our experiments. We will include this explanation in the Appendix of the revised paper.
>
>
> **Is the $f_w$ in Eq 2 and Eq 4 shared (similar with $b_m$)? If so, can you elaborate on why they are shared?**
>
> The weights are indeed shared between Eq. 2 and Eq. 4. This is required for the gating functionality to work (section 3.2): in the proof in footnote 1, the terms belonging to the out-of-phase input $z_2$ can only cancel out when the same weight $w_2$ is used for both the $\psi$ and $\chi$ terms.
>
>
> **The term "maximally distant phase" is used a few times. It would be better to include a precise definition for that.**
>
> We agree with the reviewer that this term is not well-defined in the current version of the paper. We will include the following definition in the revised version:
>
> We say that $n$ complex numbers have “maximally distant phases” if they maximize their pairwise cosine distance to one another.
>
> We will also adjust our description of Figure 4 to align with this definition by stating that the cluster centroids of the phase values belonging to the individual objects have almost maximally distant phases.
>
>
> **I didn't fully understand how to "fine-tune the decoder to reconstruct separate objects from the individual clusters". How do you know which object to reconstruct?**
>
> Since our network generally assigns the same phase values to the same object types, we manually match the reconstructions created by the separate feature vectors to the respective objects once and use the same assignment for the rest of the dataset. We will add this clarification to the revised version of the paper.
>
>
> **It is mentioned that the background cluster tends to have low magnitude. Is this general or just because the background is black? What happens when you use random sampled background (2Shapes-randBG dataset)?**
>
> Indeed, the background cluster tends to have a low magnitude when the background is black and larger magnitudes for other background colors. For the 2Shapes-randBG dataset, we observe that the model assigns larger magnitudes to the background cluster whenever the background is of a brighter gray. This is visible in the phase images in Figure 10 in the Appendix - when the input image has a black background, the phase image has a white background, indicating that the corresponding features were rescaled due to their low magnitudes. For gray backgrounds, the phase images show a blue/purple color for the background, indicating that these features were not rescaled and thus, that they had larger magnitudes. In this case, the blue/purple color corresponds to the phases that the network assigns to the background features.

---

> > ### Comment · Reviewer_NrE9 · 2022-10-22
> > **Thank you for your response**
> >
> > I am glad that the authors answered all my questions and will add clarifications to the revised version of the paper.
> > I appreciate the additional experiments on SlotAttention, which reveal some of its failure modes and can be useful to the community.
> > It is also encouraging to see that the proposed model can distinguish multiple instances of the same object shape.
> > I am happy to recommend acceptance of the paper.

---

### Decision · Action_Editors · 2022-11-07

**Recommendation:** Accept with minor revision

**Comment:**

The reviewers provided a number of constructive comments on the paper and the authors proposed some changes to address these points. The reviewers were satisfied with the proposed changes, and a unanimous 'accept' decision was arrived at. Thus, once the authors have incorporated that changes they promised to the reviewers the paper is ready for publication.

**Audience:**

TMLR's audience will likely find this paper very interesting. It provides an alternative means of segmenting and grouping features in images compared to other currently popular approaches, most notably, slot-based systems. If this can be scaled up, it would potentially provide a powerful new means of doing image segmentation. Though this paper does not show that it can be scaled up, this initial contribution demonstrates the technical soundness of this idea, and that it works in principle. It is also a broadly interesting proposal because the idea of phase-based segmentation derives from neuroscience theories.

**Claims And Evidence:**

This paper describes a new auto-encoder architecture that utilises complex-valued activation functions. The central claim of the paper is that the magnitude of the complex vectors can be used to represent features, while the phase of the vectors can be used to bind different features into distinct objects. The authors provide clear evidence for such phase-based segmentation on relatively simple visual stimuli with multiple objects (e.g. MNIST + Shapes).